

# Inter-comparison of snow depth over sea ice from multiple methods

Lu Zhou[1], Julienne Stroeve[2,3,4], Shiming Xu[1,5], Alek Petty[6,7], Rachel Tilling[6,7], Mai Winstrup[8,9], Philip Rostosky[10], Isobel R. Lawrence[11], Glen E. Liston[12], Andy Ridout[2], Michel Tsamados[2], and Vishnu Nandan[3]

[1]Ministry of Education Key Laboratory for Earth System Modeling, Department of Earth System Science, Tsinghua University, Beijing, China
[2]Centre for Polar Observation and Modelling, Earth Sciences, University College London, London, UK
[3]Centre for Earth Observation Science, University of Manitoba, Winnipeg, Canada
[4]National Snow and Ice Data Center, University of Colorado, Boulder, CO, USA
[5]University Corporation for Polar Research, Beijing, China
[6]NASA Goddard Space Flight Center, Greenbelt, MD, USA
[7]Earth System Science Interdisciplinary Center, University of Maryland, College Park, MD, USA
[8]DTU Space, Technical University of Denmark, Lyngby, Denmark
[9]Danish Meteorological Institute (DMI), Copenhagen, Denmark
[10]University of Bremen, Institute of Environmental Physics, Bremen, Germany
[11]Centre for Polar Observation and Modelling, University of Leeds, UK
[12]Colorado State University, Cooperative Institute for Research in the Atmosphere (CIRA), Fort Collins, CO, USA

**Correspondence:** Julienne Stroeve (stroeve@nsidc.org)

**Abstract.** In this study, we compare eight recently developed snow depth products that use satellite observations, modeling or a combination of satellite and modeling approaches. These products are further compared against various ground-truth observations, including those from ice mass balance buoys (IMBs), snow buoys, snow depth derived from NASA's Operation IceBridge (OIB) flights, as well as snow depth climatology from historical observations.

Large snow depth discrepancies between the different snow depth data sets are observed over the Atlantic and Canadian Arctic sectors. Among the products evaluated, the University of Washington snow depth product (UW) produces the overall deepest spring (March-April) snow packs, while the snow product from the Danish Meteorological Institute (DMI) provide the shallowest spring snow depths. There is no significant trend in the mean snow depth among all snow products since the 2000s, despite the great differences in regional snow depth. Two products, SnowModel-LG and the NASA Eulerian Snow on Sea Ice

Model (NESOSIM), also provide estimates of snow density. Arctic-wide, these density products show the expected seasonal evolution with varying inter-annual variability, and no significant trend since the 2000s. The snow density in SnowModel-LG is generally higher than climatology, whereas NESOSIM density is generally lower. Both SnowModel-LG and NESOSIM densities have a larger seasonal change than climatology.

      Inconsistencies in the reconstructed snow parameters among the products, as well as differences between in-situ and airborne

observations can in part be attributed to differences in effective footprint and spatial/temporal coverage, as well as insufficient observations for validation/bias adjustments. Our results highlight the need for more targeted Arctic surveys over different spatial and temporal scales to allow for a more systematic comparison and fusion of airborne, in-situ and remote sensing observations.



# 1  Introduction

Snow on sea ice plays an important role in the Arctic climate system. Snow provides freshwater for melt pond development and when the melt ponds drain, freshwater to the upper ocean (Eicken et al., 2004). In winter, snow insulates the underlying sea ice cover, reducing heat flux from the ice-ocean interface to the atmosphere and slowing winter sea ice growth (Sturm and Massom, 2017). Snow also strongly reflects incoming solar radiation, impacting the surface energy balance and under-ice algae and phytoplankton growth (Mundy et al., 2009). Furthermore, sea ice thickness cannot be retrieved from either laser or
radar satellite altimetry without good knowledge of both the snow depth and snow density (Giles et al., 2008; Kwok, 2010; Zygmuntowska et al., 2014).

Despite its recognized importance, snow depth and density over the Arctic Ocean remains poorly known. Most of our understanding comes from measurements collected from Soviet North Pole drifting stations and limited field campaigns. The Warren et al. (1999) climatology and the recently updated Shalina and Sandven (2018) climatology, hereafter $W99$ and $SS18$,
respectively, have provided a basic understanding of the seasonally- and spatially-varying snow depth distribution over Arctic sea ice. However, these data were collected from stations between 1950 and 1991 and were largely confined to multi-year ice (MYI) in the central Arctic. Hence, they are not representative of the Arctic-wide snow cover characteristics of recent years (Webster et al., 2014). Today, the Arctic Ocean has largely transitioned from a regime dominated by thicker, older MYI to one dominated by thinner and younger first-year ice (FYI) (Maslanik et al., 2007, 2011). In addition, the length of the ice-
free season has increased (delayed freeze-up and early melt-onset/break up), reducing the amount of time over which snow can accumulate on the sea ice (Stroeve and Notz, 2018). In response to this data gap, several groups are working to produce updated assessments of snow on sea ice using a variety of techniques.

From satellites, several studies have attempted to use passive microwave brightness temperatures to retrieve snow depth, on FYI (Markus et al., 2011) and MYI (Rostosky et al., 2018; Kilic et al., 2019; Braakmann-Folgmann and Donlon, 2019;
Winstrup et al., 2019). Other studies have attempted to model snow accumulation over sea ice using various atmospheric reanalysis products as input (Blanchard-Wrigglesworth et al., 2018; Petty et al., 2018; Liston et al., 2019; Tilling et al., In preparation). Some promise has also been shown in mapping snow depth by combining satellite-derived radar freeboards from two different radar altimeters (Lawrence et al., 2018; Guerreiro et al., 2016) and from active - passive microwave satellite synergies (Xu et al., In preparation). With the launch of ICESat-2, additional possibilities exist to combine radar and lidar
altimeters to directly retrieve snow depth (Kwok and Markus, 2018). Such approaches are paving the way for proposed future satellite missions (i.e., ESA's CRISTAL (Kern et al., 2019)).

In recognition of the numerous new snow data products available, it is timely to provide an inter-comparison of these products so that recommendations can be made to the science community as to which data product best suits their needs. Towards this end, we provide a comprehensive intercomparison between eight new snow depth products and evaluate them against various
in situ observations and different NASA's Operation IceBridge (OIB) snow depth products. Since these datasets do not have common spatio-temporal resolutions, we limit our comparisons to monthly averages between October-November (from now





referred to as the autumn period) and March-April (spring period) from 2000 to 2018 and also limit our region to the Arctic basin (i.e., we exclude regions such as the Sea of Okhotsk, Bering Sea, Baffin Bay/Davis Strait, and the East Greenland Sea).

The paper is organized as follows. The next section describes the details of each snow product, snow depth observations and
climatology data used for comparison. Comparisons among snow products and between observations are shown in Section 3. Snow density products are discussed in Section 4. Implications for further snow observations are discussed in Section 5. And the conclusion is in Section 6

# 2 Data and methods

In this section, we introduce: (1) observational datasets of snow depth used for validation/comparison, (2) climatological
snow depth products, (3) passive and active microwave snow depth products and (4) reconstructed snow depth estimates from models. The inter-comparison among all snow products and comparison between these products and measurements are based on their own spatial and temporal resolution. However, the spatial and temporal resolution varies considerably between products. Details on resolution are provided in Table 1.

Therefore, an additional comparison was made between OIB and the different snow depth products at the coarsest spatial
$(100 \times 100 km)$ and temporal (monthly) resolution. Below we briefly describe each data set and refer the reader to the references for each data product for more detailed information on the individual algorithms.

## 2.1 Measurements used for comparison

### 2.1.1 In situ Observations

Ice Mass Balance Buoys (IMBs) are designed to provide snow depth and ice thickness information, and have generally been
deployed within undeformed MYI (Richter-Menge et al., 2006). In this study, we use snow depth measurements from IMBs deployed by US Army Cold Regions Research and Engineering Laboratory (CRREL). Each buoy is equipped with acoustic sounders above and below the ice with an accuracy of 5 mm for depth measurements (Richter-Menge et al., 2006). Following Perovich et al. (2009), snow depth measurements greater than 2 m or less than 0 m are removed. In total, 58 CRREL buoy tracks are used between 2010 and 2015 (available at: http://imb-crrel-dartmouth.org). To align the daily and monthly products together
and eliminate the effects from daily missing measurements, we average the data into monthly averages after first creating daily averages from the four-hourly observations. See, Table S1 for the listing of buoys used, along with their dates/time periods of operation.

Snow buoys have been deployed by the Alfred Wegener Institute (AWI) since 2010 (Nicolaus et al., 2017), and provide snow depth estimates from 4 separate snow depth pingers. These are averaged together to provide one snow depth value at each buoy
location. Here, we use snow depths from 28 AWI snow buoys between 2013-2017 (accessible at: http://data.meereisportal.de). These buoys are also listed in Table S1. Similar is done for the CREEL buoys, with snow depths first averaged into daily averages before monthly averages are derived.



### 2.1.2 OIB Airborne observations

Since 2009, NASA Operation IceBridge (OIB) has conducted airborne profiling of Arctic sea ice every spring since 2009,
generally across the western Arctic. Snow depth observations are derived with an ultra-wideband quicklook snow radar (Paden et al., 2014), capable of retrieving snow depth from the radar echo from both air-snow and snow-ice interfaces (Kurtz and Farrell, 2011). Snow depth can then be retrieved through retracking and compensation from the radar echogram. Despite OIB campaigns providing unique large-scale and high spatial resolution observations, the swath is limited due to the airborne nature of the measurements. Furthermore, most flights tracks cover a limited area from the north of Greenland towards Alaska and
data covers only a limited time period, namely March and April.

Several algorithms have been developed to derive snow depth from this radar system (Kwok et al., 2017). While different algorithms show general agreement in regional snow depth distributions, larger interannual variability than found in $W99$ among these algorithms and the mean differences mainly result from the detection and localization of snow-air and snow-ice interfaces from the radar returns (Kwok et al., 2017). Taking these differences into consideration, we use four OIB snow depth
products from: (1) Sea Ice Freeboard, Snow Depth, and Thickness data Quicklook product (quicklook) available from NSIDC website and examined in King et al. (2015) and Kwok et al. (2017); (2) NASA Goddard Space Flight Center (GSFC) (Kurtz et al., 2013, 2015); (3) Jet Propulsion Laboratory (JPL) (Kwok and Maksym, 2014) and; (4) snow radar layer detection (SRLD) (Koenig et al., 2016). These four data products and snow products analyzed in this paper overlap only in 2014 and 2015, and thus we limit our comparisons with OIB-derived snow depths to those two years. Each OIB product is first regridded to the
$100 \times 100 km$ polar stereographic grid, and then all daily flight tracks are averaged together to produce monthly mean OIB snow depths (March or April) for each year. Then, we also compare snow reconstructions on their native spatio-temporal resolution with OIB measurements, which is provided in the Supplementary Material.

## 2.2 Snow depth climatologies

In addition to the airborne and in-situ snow measurements, we use two types of conventional large-scale snow products that
are often used in the derivation of sea ice thickness from radar or laser altimetry (Ricker et al., 2014; Kwok and Cunningham, 2008), the Warren et al. (1999) ($W99$) and the Shalina and Sandven (2018) ($SS18$) climatologies. $W99$ provides distributions of snow depth and density for each calendar month by assembling "North Pole" (NP) drifting station observations from the 1950s to 1990s. A two-dimensional quadratic function is adopted to fit the measurements to the Arctic basin. $W99$ also provides a climatology of snow water equivalent (SWE) from January to December. This is derived using snow depths and densities
measured along snow lines, and if unavailable, Arctic-mean density for that month is used. Like snow depth, a two-dimensional quadratic fit was applied to the SWE data [7].

$SS18$ combines NP data (as in $W99$) with additional snow data from the Soviet airborne expeditions (Sever), to produce spring (March-April-May) snow depth fields. Since the aircraft would land on level FYI, $SS18$ is not limited to MYI in the central Arctic (as $W99$), but includes FYI in the Eurasian seas as well (Shalina and Sandven, 2018). The spatial resolution of
the $SS18$ climatology is $100 \times 100 km$ within Arctic basin.



## 2.3 Satellite- and model-based snow depth products

Eight snow depth data sets are included in this inter-comparison study. They mainly fall into 2 categories: (1) snow reconstruction using atmospheric reanalysis data as input to a snow accumulation model together with snow redistribution by sea ice drift; (2) snow depth retrieved from satellite data, including passive microwave-based snow retrieval, blended satellite-derived radar

sea ice freeboards at two different frequencies, and active-passive satellite (combining CryoSat-2 and SMOS) data synergy. Here, snow depth is defined as the average thickness of snow over the entire grid-cell area, not just over the sea ice-covered fraction.

    The first category includes four different new products: the distributed snow evolution model (SnowModel-LG) (Liston et al., 2019; Stroeve et al., 2019); NASA Eulerian Snow on Sea Ice Model (NESOSIM) (Petty et al., 2018); the Centre for Polar

Observation and Modelling (CPOM) model (Tilling et al., In preparation); and the Lagrangian Ice Tracking System for snowfall over sea ice from University of Washington (UW) (Blanchard-Wrigglesworth et al., 2018). The second category includes the following snow depth products: the products from the University of Bremen (PMW Bremen) (Rostosky et al., 2018) and the Danish Meteorological Institute (PMW DMI) rely on satellite passive microwaves at multiple frequencies/polarizations for their snow depth retrieval algorithms (Winstrup et al., 2019). The dual-altimeter snow thickness (DuST) product (Lawrence

et al., 2018) is derived from combining data from the CryoSat-2 (Ku-band) and AltiKa (Ka-band) satellite radar altimeters. The DuST product also combines Envisat (radar altimeter: Ku-band) and ICESat (laser altimeter) data to retrieve snow depth during their period of overlap. Finally, Department of Earth System Science, Tsinghua University (DESS) combines Ku-band altimeter (CryoSat-2) and L-band passive microwave radiometer (SMOS) to retrieve snow depth based on two physical models (Xu et al., In preparation). Each approach uses vastly different methodologies that are discussed in more detail below and

summarized in Table 1.

### 2.3.1 Reanalysis-based snow depth reconstruction

**SnowModel-LG**

SnowModel-LG is a prognostic snow model originally developed for terrestrial snow applications, now adapted for snow depth reconstruction over sea ice using Lagrangian ice parcel tracking (Liston et al., 2019). Physical snow processes are included, such as blowing snow redistribution and sublimation, density evolution and snow pack metamorphosis. SnowModel-LG is used

in a Lagrangian framework to redistribute snow around the Arctic basin as ice moves. Tracking begins on August 1st in 1980 assuming snow-free initial condition, and accumulates snow until July 31st of the next year. On July 31st, any remaining snow that has becomes isothermal and saturated with meltwater becomes superimposed ice and is no longer identified as 'snow'. This is the only data product that includes snow depth during the melt season since 1980s. The essential inputs to this data

product are atmospheric reanalysis estimates of precipitation, 2m air temperature, wind speed and direction, and weekly ice motion vectors from NISDC (Stroeve et al., 2019). Weekly ice motion vectors are linearly interpolated to daily resolution. Output are snow depth, snowfall amount, rainfall amount, snow water equivalent (SWE) and bulk snow density (Liston et al., 2019), and are provided on a $25 \times 25 km$ EASE grid. A recent study (Stroeve et al., 2019) implemented SnowModel-LG using





NASA MERRA2 atmospheric reanalysis fields (Gelaro et al., 2017). They found that the model tended to underestimate the
snow depth in comparison to OIB quicklook data product (Kurtz et al.). While the differences between SnowModel-LG snow
depths and those from OIB varied from year-to-year, mean scaling factors (spatially constant but varying from year-to-year)
were derived to bias-correct the modeled snow depths to match those from OIB over the full 1980-2018 simulation, while
preserving the spatial variability of differences. The temporal resolution of SnowModel-LG is daily between 1st August 1980
and 31st July 2018.

**NESOSIM**

The NASA Eulerian Snow On Sea Ice Model (NESOSIM) is a three-dimensional, two-layer (vertical), Eulerian snow budget
model (Petty et al., 2018). NESOSIM includes two snow layers: old compacted snow and new fresh snow. Wind-packing and
snow loss to the leads are included, and were used to calibrate NESOSIM with historical snow depth and density observations
from the NP drifting station data. NESOSIM was run using several atmospheric reanalyses, including ERA-I (Dee et al., 2011),
JRA55 (Ebita et al., 2011) and MERRA (Rienecker et al., 2011) and a median of these daily reanalysis snowfall estimates is
used in this study. The model is also forced with near-surface wind fields from ERA-I, NSIDC Polar Pathfinder sea ice drift
vectors and Bootstrap passive microwave ice concentrations. Snow accumulation is initialized at the end of summer (default
of August 15th) and run until the following spring (May 1st). To initialize snow depth in mid-August, NESOSIM linearly
scales the August snow depth in the $W99$ climatology based on the ratio between duration of the summer melt season and
the climatological summer duration. The duration of the summer melt season is defined based on ERA-I air temperatures,
and climatological summer melt duration is from Radionov et al. (1997). Snow depth is further equally divided into the 'old'
and 'new' snow layers, with snow transferred from the 'new' to the 'old' snow layer based on the wind conditions. Snow
is then accumulated and evolves dynamically with sea ice motion through a divergence-convergence and an advection term.
Daily snow depth (mean depth over the sea ice fraction) and snow density and snow volume (per unit grid-cell) are available
from August 15th to April 30th for each year, at a spatial resolution of $100 \times 100 km$ in a polar stereographic grid. NESOSIM
provides the effective snow depth of the ice covered fraction, and the mean grid-cell snow depth. In order to be consistence
with snow depth estimates from the other methods, we will use the latter in this study.

**CPOM**

The CPOM snow depth product (Tilling et al., In preparation) is initialized on a Lagrangian grid with a spacing of $10 km$
running from $40°$N to the pole. Snow accumulation begins on 15th August each year using the $W99$ August snow depth and
a fixed density of 350 $kg/m^3$ on all ice covered (sea ice concentration > 15%) grid points. This initial snow layer is kept
separated from any accumulated snow after the model has started running. The model then steps daily through the winter,
accumulating snow in SWE. Snow parcels are moved using NSIDC Polar Pathfinder ice motion data, and any parcels moving
outside the ice-covered region are removed. New parcels covered by expansion of the ice-covered region become active with
no initial snow. Where the $2m$ ERA-I air temperature is above freezing point, the daily ERA-I SWE of snowfall is added





to the already accumulated column. A fraction of the accumulated snow is removed when the wind speed exceeds $5ms^{-1}$ using a function proportional to wind speed and lead fraction (see (Schröder et al., 2019), table 1). Finally, the total column of accumulated SWE at each Lagrangian point is converted to snow depth using a daily snow density function constructed in a similar way to Kwok and Cunningham (2008), which is added to the initial snow layer (if present). The irregularly spaced

snow data from the Lagrangian grid are re-gridded onto a regular $10km^2$ Polar stereographic projection using an averaging radius of $50km$ to give a snow depth map for each day of winter.

**UW**

The last reanalysis-based snow reconstruction is from the University of Washington (UW). In general, the algorithm accumulates cold-season snowfall along sea-ice drift trajectories using 12-hourly snowfall from ERA-I, weekly NSIDC sea ice

vectors and weekly-averaged NOAA-NSIDC sea ice concentration (Meier et al., 2013). A Lagrangian Ice Tracking System (LITS) (DeRepentigny et al., 2016) is used to backward track each grid point from the first week of April to the last week of the previous August (Blanchard-Wrigglesworth et al., 2018). This tracking system has a claimed accuracy of  $50km$ after 6 months of tracking (DeRepentigny et al., 2016). Once the 6-month trajectories are established for each ice parcel, the algorithm accumulates weekly averaged snowfall along each parcel trajectory. A sea ice concentration correction is further imposed every

week (one time-step). Specifically, if the ice concentration drops below 15%, the accumulation stops and the trajectory is ended at the previous time-step.

Only monthly snow depths in April are available for the period from 1980 to 2015. Spatial resolution of the data set is $75 \times 75km$ on a polar stereographic grid.

### 2.3.2    Satellite-based snow depth retrieval

**DuST**

Lawrence et al. (2018) derives snow depth by utilizing the difference in freeboards retrieved from satellites operating at different frequencies. Specifically, satellite data from ESA's CryoSat-2 (CS-2, Ku-band radar satellite altimeter operational since 2010) and CNES/ISRO's Altika (Ka-band radar satellite altimeter, 2013-present) are used. The deviation of each satellite's return from its "expected" dominant scattering horizon (the snow surface for Ka-band and the ice/snow interface for Ku-band) is

quantified using independent snow and ice freeboards from OIB. Using a spatially variable correction function, AltiKa and CS-2 freeboards are calibrated to the snow surface and snow/ice interfaces respectively, allowing snow depth to be estimated as the difference between the two. A caveat to the approach is that since OIB data are only available during March/April and cover limited regions of the Arctic, the calibration of the CS-2 and AltiKa freeboards with OIB may not be valid during other months and/or regions. The same methodology has also been applied to ICESat and Envisat satellites, whose active periods

overlap between 2003 and 2009, and these data are also included.

This Dual-altimeter Snow Thickness (DuST) product produces: (1) monthly snow depth maps from October to April during CS-2/Altika period (2013-present), and (2) bi-monthly annual snow depth maps (March-April or October-November) during





the ICESat-Envisat period (2003-2008). For both time-periods, the snow depth is gridded on a $1.5°$ longitude $\times 0.5°$ latitude grid. The data is limited to below $81.5°$N due to the upper latitude of AltiKa/Envisat.

**PMW Bremen**

The University of Bremen's PMW snow depth product described in Rostosky et al. (2018) on FYI is available during the AMSR-E/2 period and the algorithm is adapted from Markus and Cavalieri (1998), which derived from a series of passive microwave sensors, such as the Scanning Multichannel Microwave Radiometer (SMMR) (from 1979), and continuing on through the Special Sensor Microwave Imager (SSM/I) and SSM/I Sounders (SSMIS). The latter algorithm computes the spectral gradient ratio between 18.7 and 37 $GHz$ vertical polarization brightness temperatures (Tbs) to generate 5-day averaged snow depth estimations over FYI (Markus et al., 2011). This algorithm has limitations for wet snow springand multiyear (Comiso et al., 2003) and large sensitivity to surface roughness (Stroeve et al., 2006). Snow depths over smooth FYI were found to be accurate in comparisons with OIB during 2009 and 2011 ($RMSE < 0.06$ m over a shallow snow cover), while significant biases were found over rougher FYI or MYI (Brucker and Markus, 2013). The spatial resolution is similar to typical passive microwave measurements, at $25 \times 25 km$ in the polar stereographic grid. Rostosky et al. (2018) further extends the approach of Markus and Cavalieri (1998) to also include snow depth over MYI using the lower frequencies of 6.9 $GHz$ from the NASA Advanced Microwave Scanning Radiometer AMSR-E and the JAXA Global Change Observation Mission-Water (GCOM-W) AMSR2 instrument. We refer to this method as the PMW Bremen snow depth product. The gradient ratio between vertical polarized brightness temperatures (Tbs) at 18.7 $GHz$ and 6.9 $GHz$ helps to mitigate the retrieval sensitivity problems over MYI during March and April. Specifically, robust linear regressions were derived based on fitting 5 years' (2009, 2010, 2011, 2014, and 2015) NOAA Wavelet ("WAV") Airborne Snow Radar-Snow Depth on Arctic Sea Ice Data Set (Newman et al., 2014) and the polarized Tb gradient ratio. Snow depths over different ice types were fitted separately, using OSI-SAF sea ice type map to distinguish between FYI and MYI. No evaluation/validation were performed in regions outside of OIB measurements. Daily snow depth maps with spatial resolution of $25 \times 25 km$ on the polar stereographic grid for winter months from November to April since 2002 are available. It should be noted that snow depth over MYI is only available during March and April when the OIB data were available to constrain the model. Snow depth uncertainty is estimated to be between 0.1 and 6.0$cm$ over FYI and between 3.4 and 9.4$cm$ over MYI.

**PMW DMI**

In Winstrup et al. (2019), snow depth is derived by a random forest regression model based on passive microwave Tbs from AMSR-E and AMSR2. The model was trained using a Round Robin Data Package (Pedersen et al., 2018), with OIB campaigns snow thicknesses provided by NSIDC including IDCSI4 and quicklook products and collocated brightness temperatures. Training was performed using OIB data from the period April 2009- March 2014, leaving the remaining OIB data for validation purposes. Specifically, multi-channel Tbs ranging from C-band (6.9 $GHz$) to 89 $GHz$ are adopted, both vertical and horizontal polarization. The random forest consists of 500 regression trees, each derived from bootstrapping the input data, and a





maximum limit of five features for each leaf in the regression trees. Derived snow depths were found to be in good agreement
       with the OIB data retained for validation. The passive microwave snow depth product from the Danish Meteorological Institute
       (PMW DMI) constructs spring-time (March and April) daily snow depth throughout the 2013 and 2018 at $25 \times 25km$ spatial
       resolution on the EASE-Grid 2.0 (Brodzik et al., 2012).

**DESS**

The Department of Earth System Science in Tsinghua University (DESS) algorithm retrieves both sea ice thickness and snow
       depth simultaneously by using sea ice freeboard from CS2 and L-band (1.4 $GHz$) Tbs from the Soil Moisture and Ocean
       Salinity (SMOS) satellite (Xu et al., In preparation). The active period of these two satellites are both from 2010 to the present.
       Specifically, this algorithm combines a hydrostatic equilibrium model and improved L-band radiation model. Sea ice freeboard
       is converted to sea ice thickness based on hydrostatic equilibrium (Laxon et al., 2003), using assumptions on snow, ice and
water densities. Tbs from SMOS can be used to retrieve thin sea ice thickness (Kaleschke et al., 2012) and snow depth over thick
       ice (Maaß et al., 2013). The L-band radiation model is further improved by adding vertical structure of temperature and salinity
       in sea ice and snow (Zhou et al., 2017). In order to obtain the missing measurements resulting from limited upper latitude in
       the SMOS satellite, L-band Tbs spanning inclination angles from $0°$ to $40°$ and from $85°$N to $87.5°$N is approximated using
       Tbs of all frequencies in AMSR-E and AMSR2 through a back-propagation machine learning process. By combining the two
observational datasets, the uncertainty in both sea ice thickness and snow depth can be reduced. Unlike optimal interpolation
       based sea ice thickness synergy in Ricker et al. (2017), the uncertainty in ice thickness is reduced through an explicitly retrieved
       snow depth.

       Both sea ice thickness and snow depth are available in the DESS product. Here we use the monthly snow depth maps
       available for March of each year since 2011, which are provided at a spatial resolution of $12.5 \times 12.5km$ on a polar stereographic
grid.

## 3    Snow depth inter-comparison

In this section, we first compare snow depths among the eight snow products (Section 3.1), and then we assess how these snow
products compare against various observations (Sections 3.2 and 3.3), and the climatologies (Section 3.4). Apart from entire
common region comparisons (Figure S1), we also explore consistency and mean state of all products over different sub-regions
in the Arctic: the Canadian Arctic sector (CA) including Canadian Archipelago, Atlantic (Atlantic) and Pacific & Central Arctic
       (Pacific) sectors (see Figure S1). Over these regions, sea ice conditions vary considerably, with mostly thick MYI within the
       CA, and thinner FYI elsewhere. Also, precipitation patterns differ between these regions with more precipitation falling over
       the Atlantic sector as a result of the proximity to the North Atlantic storm tracks. Given the relatively long time period and
       large spatial coverage provided by SnowModel-LG, this snow depth product is used as the reference product when comparing
regional consistency between products. To align the temporal and spatial resolution between all snow products, comparisons
       are carried out after 2000 and focused on the early and late winter months of October/November and March/April.





## 3.1 Comparison among snow products

Spatial maps of monthly snow depths across all the data products, as well as the $W99$ and $SS18$ climatologies, are shown in Figure 1 and 2 for the autumn and spring periods in 2014, respectively (on native grid projections). The spatial patterns in all products are in broad agreement that thicker snow occurs north of Greenland and the CA sector and thinner snow in the seasonal ice zones (i.e., Baffin Bay). Some products (i.e., NESOSIM) also capture thicker snow in the East Greenland Sea and the Atlantic sector. However, large differences are apparent in mean snow depths magnitudes, and regional discrepancies are apparent. In particular, NESOSIM indicates that the thickest spring snow occurs in the East Greenland Sea, while in DESS, the deepest snow is concentrated in the Canadian Arctic sector.

During the autumn period, the average snow thickness is respectively $2.0cm$ and $9.0cm$ thicker in NESOSIM and DuST than in CPOM, when compared over the common region (Figure S1) of all datasets. MERRA2-based SnowModel-LG runs show a similar spatial pattern as NESOSIM and CPOM but shallower snow than NESOSIM and slightly deeper snow depths than CPOM in the north of Greenland and Svalbard. They share similar snow patterns with $W99$, but with $15.0cm$ shallower snow on average. Among all snow depth products shown in Figure 1, DuST shows the thickest snow pack among snow products ($16.0cm$ mean snow), though the spatial coverage is limited.

During spring, average snow depths from the eight products range from about $25.0cm$ to $30.0cm$ within the Arctic common region (Figure S1). Deeper snow exists within the East Greenland Sea for SnowModel-LG, NESOSIM and UW. DESS shows the thickest snow pack over the CA. All reanalysis-based products except for CPOM show thicker snow cover over the North Atlantic sector resulting from the North Atlantic storm tracks. In $W99$ and $SS18$, snow is also deepest (over $35.0cm$) north of Svalbard. NESOSIM further suggests thick snow over Davis Strait, with spring averaged snow depths greater than $25.0cm$. This is in stark contrast to the other data sets over the FYI in that region.

The histogram of average snow depth among these products are shown in Figure 3 for the period 2000 to 2018 during autumn and spring respectively. Since the DuST product is limited to below $81.5°$N, histograms of snow depth distribution are limited to the regions up to $81.5°$N. Out of all the reanalysis-based data products, snow depth distributions in NESOSIM are shifted towards slightly deeper snow packs ($8.0cm$) than those from SnowModel-LG ($7.0cm$) and CPOM ($6.0cm$) during autumn. The modal and distribution of snow are similar in SnowModel-LG, NESOSIM and CPOM. The deepest snow packs during October/November are found in DuST, with mode of the distribution being $\sim 17.0cm$, considerably larger than the other products. Since the region sampled is the mixture of MYI and FYI for latitudes below $81.5°$N, snow depths in DuST are larger than expected.

During spring, however, DuST snow depth distributions closely resemble those from PMW Bremen, with mean snow depth values around $19.0cm$. Thus, in DuST, there is a very small seasonal cycle in snow accumulation (seen also in Figure 4). PMW DMI exhibits the overall smallest ($12.0cm$) mean spring snow depths while UW shows the largest ($23.0cm$). Mean snow depth in NESOSIM is $2.0cm$ higher than in SnowModel-LG. According to DESS, mode of the snow thickness peaks at $20.0cm$ during March/April. Conclusions are similar if we omit DuST and extend the analysis up to $87.5°$N (region shown in Figure S1b), but the bimodal snow depth distribution in autumn is more obvious, representing less snow over newly formed sea ice





areas, and deeper snow over MYI. As expected, snow depths shift to overall deeper snow packs in spring when including the higher latitudes (Figure S2).

We additionally examine snow differences over the three different sectors in spring 2015 (Figure S3). For this comparison, each data product is aligned with SnowModel-LG: solid lines show results from SnowModel-LG, while dashed lines indicate

the various other products. All the reanalysis-based snow depth products show the thickest snow over the North Atlantic. In contrast, the satellite-based products indicate more snow accumulating over CA. While this is only one year of comparison, it shows that regional differences in snow accumulation can be quite pronounced depending on data set used.

Time-series (2000-2018) of monthly mean snow depths from September to April averaged over regions up to $81.5°$N are displayed in Figure 4 including the snow climatology from $W99$ (corresponding results up to $87.5°$N are shown in Figure

S4). Considerable differences in snow depth magnitudes are found among the different products though all reanalysis-based snow reconstructions, namely SnowModel-LG, CPOM, NESOSIM and UW have similar inter-annual variability in April since 2000 (see more details Table S2). DESS exhibits similar inter-annual variability as the reanalysis-based products ($R^2$= 0.42 to SnowModel-LG; 0.68 to NESOSIM and; 0.32 to CPOM). However, inter-annual variability in PMW Bremen and DuST are not consistent with the reanalysis-derived snow depths.

By April, NESOSIM has accumulated more snow than SnowModel-LG, especially after 2012. However, it should be noted that the winter time snow accumulation is largest in SnowModel-LG due to lower snow initial conditions in September. The lowest snow accumulation of the reanalysis-based snow depth products is found in the CPOM data set, while UW holds the most snow at the end of freezing season in most years. The climatological snow depths of $W99$ on the other hand show the largest seasonal range in snow accumulation, and the deepest snowpacks in March, the month when snow reaches its maximum

depth.

Overall, the inter-annual variability of monthly snow depth as averaged for all regions up to $81.5°$N in Figure 4 is small among all snow products, ranging from $2.0cm$ in November to about $2.0cm$ to $3.0cm$ in April. This is also true for averages up to $87.5°$N (Figure S5) and is about half of that previously observed from climatology snow depth data products (Warren et al., 1999), where inter-annual variability was estimated to be about $4.3cm$ in November and $6.1cm$ in April. Note however,

the climatological estimation of inter-annual variability included snow depth uncertainty. It should be noted that DuST shows a significant positive snow depth bias from the Envisat period through CryoSat-2 period, but this could be due to the limitation in processing the laser altimeter freeboard in the product. This feature is also not observed in the passive-microwave based snow products (e.g., PMW Bremen), nor is it observed in DESS (time-series in DMI is too short to be assessed).

We do not find a significant trend in most of the snow products since 2000. The reanalysis-based snow products offer the

possibility to assess longer term trends (i.e., since 1980), while those from passive microwave at a similar temporal time-scale are limited to FYI regions only. Figure 5 shows snow depth trends from 1991 to 2015 (common period for reanalysis-based products) for SnowModel-LG, CPOM and UW. During spring, all three products show statistically significant positive trends north of Greenland and the Canadian Archipelago. Spring trends within the rest of the Arctic basin are mostly negative, but only statistically significant for SnowModel-LG, and for CPOM over the Barents Sea. In autumn, the area with statistically

significant negative snow depth trends from SnowModel-LG is extended, which is likely a result of delays in freeze-up (Markus



et al., 2009; Stroeve and Notz, 2018). CPOM also shows negative trends in these regions, but not as large as those simulated by SnowModel-LG. Spring and autumn mean snow depth trends in each year from SnowModel-LG as computed over the entire Arctic basin are -0.5$cm/decade$ and -0.9$cm/decade$, respectively, although some regions show larger trends. In CPOM, negative trends (-0.47$cm/decade$) are limited to autumn.

## 350   3.2   Comparison against OIB

Next, we assess the snow products against the four different OIB-derived snow depth estimates. We first compare OIB and snow products after gridding both to $100 \times 100km$ and evaluating the monthly averages Taking the quicklook product as an example, there are about 1,300 OIB mean measurements per 100 km grid cell. It should be noted that snow depths from DuST, PMW Bremen and DMI are directly fitted against OIB snow depths, and as a result, these data show high correlations with

OIB, which should not be taken as a real validation (or comparison) for these products. Except for NESOSIM and UW, other reanalysis-based products are also to some extent indirectly tuned by OIB snow depths in some years. Figure 6 summarizes these results as scatter plots between the different OIB snow depth products and the various snow products in 2014 and 2015 (different colors correspond to different OIB snow depth data sets). The corresponding $R^2$ and $RMSE$ of these comparisons are shown in Table 2.

A key point from these comparisons is that while we provide $R^2$ and $RMSE$, there is no real "true" OIB observation, and thus caution is warranted when interpreting the results. In fact, a strong dependence of our validation results to the OIB data product used is evident by the different linear fitting slopes shown in Figure 6, as well as the $R^2$ values, $RMSE$ and normalised $RMSE$ ($NRMSE$: $RMSE/(max - min)$) in Table 2. Not surprisingly, the fit is best for the PMW Bremen and PMW DMI data, followed by the CPOM and DESS products, though this depends on which OIB data set is used for evaluation.

For example, CPOM performs best against SRLD ($R^2$=0.61) and worst against GSFC ($R^2$=0.43); DESS also performs best against SRLD ($R^2$=0.59) but worst against the quicklook ($R^2$=0.26). SnowModel-LG and NESOSIM have similar $R^2$ ranging from a low $R^2$ of 0.27 and 0.29, respectively against the quicklook product to a high of 0.47 (SnowModel-LG vs. SRLD) and 0.39 (NESOSIM vs. GSFC). The UW snow product on the other hand performs poorly against all OIB snow depth estimates (maximum $R^2$ of 0.31). Among all indirectly OIB fitting snow products (SnowModel-LG, CPOM, DESS, UW and NESOSIM),

$RMSE$ in CPOM is the lowest, while in DESS the $RMSE$ is over 10.0$cm$. $NRMSE$ is also computed to remove the effects of sample range. DESS shows the largest $NRMSE$ among all products. The distribution/variability in UW is quite narrow compared with other products, however, the $NRMSE$ in UW are comparable with CPOM, PMW Bremen and PMW DMI , despite of its lowest $RMSE$. High variability in OIB snow depth results from small-scales measurements, making these comparisons challenging (Sturm et al., 2002). Representative issues will be discussed further in Section 5.

We also examined the relationship for each of the data products' native grid and native temporal resolution, to see how results are influenced by grid resolution and the monthly averaging (Figure S5). The $R^2$, $RMSE$ and $NRMSE$ for each comparison are listed in Table S3. In contrast to the coarser resolution comparisons shown in Figure 6 and Table 2, more outliers, lower $R^2$ and larger $RMSE$ are found in native spatial-temporal resolution comparison (Figure S5), though again



this depends on OIB data product, and in some instances the fit (correlation) improves. There is still no significant correlation
between UW snow depths and those from OIB, but the overall $RMSE$ is the lowest recorded at less than $4.0cm$. PMW DMI
has the lowest $NRMSE$, followed closely by all other snow products, with the largest $NRMSE$ in DESS. Along with the
monthly snow products in Figure S5, Table S3 and coarse comparison in Figure 6, we can discuss how the spatial resolution
affects OIB comparison. All snow products except NESOSIM and UW show higher $R^2$ under spatial coarser resolution. These
two products, however, suggest higher $R^2$ under native spatial resolution in some OIB products comparison. Besides, the
comparisons between monthly and daily give us the hint that temporal resolution exert minor influences in OIB comparison
due to less changes in $R^2$ and $RMSE$.

Given how sensitive these results are to the OIB data sets chosen, it is impossible to conclude which snow product performs
best overall, but clearly the snow products that have been produced through tuning with OIB data show higher $R^2$ and smaller
$RMSE$s. This also points to the need for further independent observations also of more recent OIB data.

## 3.3 Comparison with buoy data

Here we further explore how well the snow products represent the temporal evolution of snow depth observed from CRREL
IMBs and AWI snow buoys. As discussed in Section 2.1.1, 86 buoy tracks (58 tracks are from CRREL and 28 tracks are from
AWI) are processed during the period of 2000 and 2017 (Table S1). Since we do not expect any of the coarsely ($100km$)
resampled products to match the local-scale of the buoy data, we start by showing scatterplots between the monthly mean
(March and April) snow depths from the buoys vs. those from the products in native spatial resolution in Figure 7. DuST is
excluded due to lack of buoy samples corresponding to the limited spatial coverage of this data product Correlations are small
for all snow products (less than 0.2), and the PMW Bremen and PMW DMI products show essentially no variability compared
to the buoy data.

Next, we compare snow accumulation along the track of each buoy and summarize as scatter plots of total snow accumulation
from early winter to spring next year (Figure 8). Specifically, the three daily-resolution products (SnowModel-LG, NESOSIM
and CPOM) are interpolated daily into the geolocation of buoys. Only buoys which survive from at least October until the
following February are considered. Snow accumulation is then calculated by subtracting the mean snow depth within first
seven days and snow state in last seven days according. In this way, the total snow increment from each buoy is determined.
Figure 8 shows the snow accumulation comparison between these daily products and buoys. None of these products show
significant correlation with buoy accumulation because several outliers weaken the overall correlation.

## 3.4 Comparison with climatology

Here we focus on how snow depth has changed over the years according to the new data products in relation to the $W99$ and
$SS18$ climatologies. Figure 9 and 10 summarize the distribution of snow in $W99$ and snow products within the Arctic basin
in autumn and spring respectively for different time-periods depending on availability of the snow products. For example,
SnowModel-LG results are summarized for 1980-2000 and 2000-2018 whereas NESOSIM is only available from 2000 to





2017. Comparison of time-periods allows for an assessment of how snow depth changed over time and how well different time-periods compare against climatology.

Both SnowModel-LG and NESOSIM show bimodal snow distribution during autumn that is not seen in the $W99$ climatology; According to SnowModel-LG, this snow thickness in the period 2000-2018 is about $3.0cm$ larger than in the two earlier
decades (1980-2000), though it is not as deep as in NESOSIM for approximately the same time period. SnowModel-LG simulations begin with snow-free conditions in August 1980, while NESOSIM initialises with a modified $W99$ snow depth, and therefore it is expected that NESOSIM will have overall deeper snow packs in October/November.

In contrast to an overall pan-Arctic increase in snow depth from SnowModel-LG, CPOM suggests that autumn snow depth has decreased by about $2.0cm$ between the 1990s and 2000s. On the other hand, an increase over time is also loosely suggested
by the DuST snow depth estimates, as DuST increases from $12.0cm$ snow depth over the ICESat period (2003-2008) to $17.0cm$ during the CryoSat-2 (2013-2018) period. However such a large increase in snow depth seems unlikely given the loss of multiyear ice in the last decade. Snow in PMW Bremen, which shows the highest snow in early winter, is $4.0cm$ thinner than $W99$, which might result from a wrong classification for FYI/MYI having strong impacts on the microwave emission over MYI (Rostosky et al., 2018). Compared to $W99$, we find that snow depth in all products is about $7.0cm$ less in the 2000s
during autumn freeze-up

Consistent with Webster et al. (2014) we find that by the end of winter (March/April), snow depths in all products are considerably lower than in $W99$, where the mean snow depth is about $35.0cm$. There are no significant differences in mean snow depth in the 1980s (1990s in CPOM) and in the 2000s in CPOM and UW, while SnowModel-LG indicates an insignificant decrease over the 39 years. DuST, however, suggests that snow is increasing from 2003-2008 to 2013-2018 in Figure 10.
SnowModel-LG is the only snow product that has a large distribution of snow depths between 0 and $10.0cm$ in March/April. Compared with $W99$, the minimum difference is $10.0cm$ for UW and the maximum is over $15.0cm$ for PMW DMI.

The climatology from Shalina and Sandven (2018) provides additional detail in the marginal seas, especially over the Eurasian seas but is limited to spring snow depth estimates. Overall $SS18$ has lower snow depths in the central Arctic compared with $W99$. Figure 11 reveals that snow distribution in $SS18$ includes two modes; $18.0cm$ and $32.0cm$ respectively,
corresponding to snow over FYI and MYI. Snow in April shows slightly thicker snow packs than in March, with a change of about $1.5cm$ in most products. Generally, differences between the $SS18$ climatology and the various snow products are similar to the $W99$ comparisons, but these comparisons provide more detail on snow over FYI.

Overall, the comparison of the different snow products against these two climatological datasets all reach a similar conclusion:: that current snow depths have decreased with respect to the climatologies. The reduction in early winter is smaller than at
the end of the winter, which implies either that the intensity of snow accumulation is weakening or that the snow accumulation period has shortened. However, caution is needed since there are large uncertainties in the snow climatology data sets and the interannual variability in these data sets is larger than in the snow products.





## 4 Snow density comparison

Only two products, SnowModel-LG and NESOSIM provide daily snow density together with their snow depths. Since the
$W99$ climatology contains both snow depth and SWE, we can compare against the $W99$ snow densities. Snow density in
$W99$ is limited to the Arctic basin. Figure 12 shows monthly mean snow density in November and April as averaged for
2000 to 2018 (SnowModel-LG) and 2000 to 2017 (NESOSIM). Corresponding monthly means from $W99$ are also shown.
SnowModel-LG suggests that snow is denser than in NESOSIM both in November and April. Considering that the snow
depth in SnowModel-LG is thinner than NESOSIM, it is found that the higher density in SnowModel-LG produce a broadly
450 equivalent SWE (not shown). It is worth noting that snow over the Atlantic sector, especially within the East Greenland Sea is
quite dense in SnowModel-LG, with mean density values above 370 $kg/m^3$ in November and April. In contrast, NESOSIM
has mostly smaller snow densities and considerably less spatial variability. However, in $W99$, snow is denser over Atlantic
sector in November, while in April, the snow has higher density over the Pacific sector.

Time-series of mean snow density within the Arctic basin in SnowModel-LG, NESOSIM and $W99$ changes are summarized
455 in Figure 13 from October to April. It is apparent that density in SnowModel-LG is larger than in NESOSIM especially at the
end of winter: The average SnowModel-LG snow density in April is about 340 $kg/m^3$ while NESOSIM is around 310 $kg/m^3$;
$W99$ falls between the two estimates at 320 $kg/m^3$. Seasonally, the $W99$ density increases from 280 $kg/m^3$ in October to
320 $kg/m^3$ in April, whereas SnowModel-LG, densities increase from 245 $kg/m^3$ to 350 $kg/m^3$. NESOSIM shows seasonal
changes on the order of 250 to 300 $kg/m^3$. Neither the NESOSIM nor SnowModel-LG densities suggest any long-term
changes in snow density, yet SnowModel-LG shows considerable interannual variability not present in NESOSIM or the $W99$
climatology.

## 5 Discussion

Low or no correlation between buoys measurements and these coarse-resolution snow products brings about the necessity
to discuss how best to compare data at different spatial/temporal resolutions, or how best to validate the coarser grid-scale
products using in situ or OIB observations. The typical spatial coverage of a buoy is a few meters and these are deployed on
level ice. Airborne surveys, such as OIB, feature larger spatial coverage, but still considerably much smaller than the satellite-
or reanalysis-based snow products, and they are temporally limited.

Since there are no measurements for validation that span the small-scale (typical of buoys or airborne observation) to large-
scale (typical of passive microwave or reanalysis based snow depth products) footprint size within a specific region, we carry
out a sensitivity study using quicklook OIB data to discuss the statistical representativeness (e.g., fitting slope) of different
spatial coverage. In each cell (here $37.5 \times 37.5 km$ is taken as one cell for analysis), the "true" snow depth (hereinafter referred
to as Hs) is defined as the mean depth (mean Hs) from all OIB measurements located in this region using three different
sampling strategies. Here, the "true" snow depth is taken as the target snow depth to retrieve using the satellite or reanalysis-
based snow depth products. The three sampling strategies are: (I) random re-sampling, for which 40 OIB samples are randomly





chosen within this region and used to compute the mean Hs; (II) segmentation re-sampling, for which a local OIB segment track with 40 consecutive samples is chosen and used to compute mean Hs; and (III) single re-sampling, for which a single OIB Hs measurement is chosen to represent the true snow depth. The strategy III essentially mimics a buoy observation in one day within typical passive satellite spatial resolution (i.e., limited spatial coverage). Specifically, in case of the chosen sample over extreme ice conditions such as over the ridged ice, Hs is chosen where its ice thickness must be within 1 sigma of the grid

cell ice thickness in strategy III.

Figure 14 shows snow distribution in strategy I (red lines) and strategy II (blue lines) within one cell over three different samples ((a), (b) and (c)). Besides, a typical case of the fitting in strategy II is shown in Figure 14 (d). Although the same mean snow depth, snow distribution is much narrower in strategy I and II than in original measurements, especially in random re-sampling . Figure S6 provides overall distribution of slopes of the fitting for the three different sampling strategies. In strategy

I (random re-sampling), due to smaller (40 samples) chosen OIB sample count, the slopes of the linear fitting lines are lower than 1 (around 0.91). If we limit the measurement of Hs to a local segment (strategy II), the slopes drop to about 0.73, which indicates that the representation worsens when comparing a local measurement and a relatively large counterpart. When we further limit the comparison to a single OIB sample (strategy III), the slopes further decrease to around 0.24, although there are still significant correlations. This result agrees with the buoys comparison in Section 3.3, where the fitting slopes between snow

products and buoy (Figure 7) is considerably flatter than against the OIB observations. Although statistically significant fitting is attained for the different sampling strategies, the decrease in the slopes is expected given the decrease of spatial coverage, or in other words, decreased representativeness of buoy or airborne measurements. This result also indicates that when tuning the prognostic/statistical models against airborne or in-situ measurements for reconstructing Hs, the representation issue should be considered in order to avoid over-fitting. However, the representation error of sea ice parameters urges more systematic and

targeted measurements to investigate in the future, further, the analysis about effects of representation on data assimilation or validation based on measurements are required for more sea ice parameters such as radar freeboard.

# 6 Conclusions

While we cannot conclusively point to which snow product best represents the "true" snow depth, this study does allow for improved understanding of the spatial distribution and temporal changes (mean state and inter-annual variability) of snow

over Arctic sea ice. In summary, this paper offers a detailed assessment of the various snow depth retrieval methods and how snow changes seasonally and over time. Large spatial and temporal discrepancies among the eight snow products are analyzed together with their seasonal variability and long-term trends.

The various snow depth products show about half the interannual variability in March/April than previously estimated by $W99$ ($3.0cm$ versus $6.1cm$). The snow products with the longest times-series show no trend in snow depth over time, although

they all show significantly less snow in spring than in the $W99$ climatology. Among these snow products, UW shows the largest snow depths, at the end of the freezing season, while PMW DMI suggests the shallowest snow packs at this time. It is worth noting that inter-annual variability in all reanalysis-derived product and DESS are consistent with each other, whereas





the inter-annual variabilities in DuST and PMW Bremen are inconsistent with any other product. Snow distribution in spring from DuST and PMW Bremen are similar especially during the CryoSat-2 time-period. DESS indicates more snow over the
CA while NESOSIM suggests more snow over the East Greenland Sea region. Due to the different snow pattern among all snow products, more field surveys should be included over the Pacific sector and Atlantic sector to capture the specific patterns due to the flooding of snow and ice from increased precipitation in that region. This was observed during the N-ICE2015 campaign (Gallet et al., 2017; Merkouriadi et al., 2017) with important implications on satellite measurements and retrieval algorithms proposed.

Each product is discussed thoroughly in relation to the corresponding OIB and buoy observations. For the OIB comparison, four different OIB products using multiple retrieval schemes are accounted for. Most snow reconstructions show significant correlation with OIB observations. However, it is hard to determine which is the best on account of (1) large differences among the four OIB snow products, (2) high sensitivity to different OIB products and (3) the use of OIB data for calibration during development of some of the snow depth products. In view of the number of recent scientific studies relying on OIB data
validation and parameterisation, we cannot emphasise enough the importance of improving the retrieval algorithms on OIB observations and also including more newly OIB measurements.

Higher winter time snow accumulation is found in the buoys than in reanalysis-based products (Figure 8), as well as larger accumulation variability in buoy measurements. However, more effort is required to simulate how snow accumulates locally since currently no product has the ability to model snow depth at local spatial scales. For the reanalysis-based products,
improvement in the ice drifting algorithm and snow redistribution parameterisation would result in more realistic snow accumulation representation. All snow products display relatively low correlation with snow depth measured by buoys, with the highest correlation $\sim 0.4$ between buoy and DESS. By subsampling the OIB measurements within the typical product large-scale footprint, we proposed a reason explaining the scatter and flattening of the linear regression between the large-scale products and buoy measurements. Further analysis and measurements at different spatial and temporal scale are needed to fully
understand these scaling issues. Besides, in consideration of spatial and temporal resolution of snow products, more research into how best to select optimal spatial and temporal scales for comparisons against field or airborne data is needed.

The combination of the snow depth climatology and the various snow depth products support the idea that snow depth is decreasing over the years during the whole freezing season, in November or April, although there are large uncertainties in the climatology due to spatial interpolation from limited in-situ observations. In addition, the winter time snow accumulation in
these products are much smaller than in the climatology. In contrast with the large inter-annual variability estimated from the climatology, all snow products reveal much smaller inter-annual variability both in November and April. Lastly, snow density in both SnowModel-LG and NESOSIM are quite different from $W99$, with lower density snow in November and denser in April according to SnowModel-LG. NESOSIM shows overall lower snow density than in $W99$. It is found snow density in these snow reconstructions has no distinct trend over the 19 years.

Since the snow distributions over the Atlantic sector and Canadian sector vary among the eight snow products, more observations are required particularly over these areas. Ambitious field campaigns (i.e. MOSAiC) will provide us with a better understanding of the seasonal evolution of snow on sea ice, but mostly over the central Arctic region only.



*Data availability.* The data are available from the authors upon request.

*Author contributions.* LZ and SMX contributed the DESS data. JS and GEL contributed the SnowModel-LG data. AP contributed the
NESOSIM and three different OIB data (JPL, GSFC and SRLD). RT and AR contributed the CPOM data. MW contributed the PMW DMI
data. PR contributed the PMW Bremen data. IRL and MT contributed the DuST data. LZ and JS wrote the paper and all authors contributed
to editing of the manuscript.

*Competing interests.* The authors declare that they have no conflict of interest

*Acknowledgements.* This work is partially supported by the National Key R & D Program of China under the grant number 2017YFA0603902
and the General Program of National Science Foundation of China under the grant number 41575076. This work is also partially supported
by Center for High-Performance Computing and System Simulation, Pilot National Laboratory for Marine Science and Technology (Qing-
dao). The work of P. Rostosky was funded by the Deutsche Forschungsgemeinschaft (DFG, German Research Foundation) Project Number
268020496 TRR 172, within the Transregional Collaborative Research Center ArctiC Amplification: Climate Relevant Atmospheric and
SurfaCe Processes, and Feedback Mechanisms $(AC)^3$. MT acknowledges support from the European Space Agency Project in part by
project "Polarice" under Grant ESA/AO/1-9132/17/NL/MP and in part by the project "CryoSat+ Antarctica" under Grant ESA ESA AO/1-
9156/17/I-BG.



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





**Table 1.** Summary of investigated snow reconstruction products.

| Product | Time span | Temporal resolution | Spatial resolution | Projection type | Method type | Reference |
|---|---|---|---|---|---|---|
| SnowModel-LG | 1980-2018 | All year (daily) | $25 \times 25km$ | EASE grid | Reanalysis-based | Liston et al. (2019) Stroeve et al. (2019) |
| NESOSIM | 2000-2017 | Aug to Apr (daily) | $100 \times 100km$ | Polar stereographic grid | Reanalysis-based | Petty et al. (2018) |
| CPOM | 1991-2017 | All year (daily) | $10 \times 10km$ | Polar stereographic grid | Reanalysis-based | Tilling et al. (In preparation) |
| UW | 1980-2015 | Apr (monthly) | $75 \times 75km$ | Polar stereographic grid | Reanalysis-based | Blanchard-Wrigglesworth et al. (2018) |
| DuST | 2003-2008, 2013-2018 | Bi-monthly (2013-2018) & monthly (2003-2008) | $1.5°$ longitude $\times 0.5°$ latitude | Up to $81.5°$ N | Active satellite-based | Lawrence et al. (2018) |
| DESS | 2011-2019 | Mar (monthly) | $12.5 \times 12.5km$ | Polar stereographic grid (up to $87.5°$ N) | Active & passive satellite-based | Xu et al. (In preparation) |
| PMW Bremen | 2003-2018 | Mar & Apr (Daily) | $25 \times 25km$ | Polar stereographic grid | Passive satellite-based | Rostosky et al. (2018) |
| PMW DMI | 2013-2018 | Jan to Apr (Daily) | $25 \times 25km$ | EASE grid 2.0 | Passive satellite-based | Winstrup et al. (2019) |

**Table 2.** $R^2$(in bold), $RMSE$ (left in bracket, units: $cm$) and $NRMSE$ (right in bracket) of various average monthly snow depth products in comparison with four OIB snow depth products, using $100 \times 100km$ monthly comparison.

| OIB Product | SnowModel-LG | NESOSIM | CPOM | UW | DuST | DESS | PMW Bremen | PMW DMI |
|---|---|---|---|---|---|---|---|---|
| quicklook | **0.27** | **0.29** | **0.54** | **0.00** | **0.36** | **0.26** | **0.59** | **0.54** |
| | (9.5,0.23) | (10.0,0.22) | (6.2,0.15) | (3.1,0.15) | (4.6,0.14) | (14.0,0.41) | (4.4,0.11) | (4.4,0.10) |
| GSFC | **0.30** | **0.39** | **0.43** | **0.10** | **0.30** | **0.48** | **0.56** | **0.37** |
| | (9.5,0.23) | (9.0,0.21) | (7.4,0.19) | (3.6,0.13) | (5.2,0.14) | (12.4,0.34) | (4.8,0.11) | (5.4,0.13) |
| JPL | **0.41** | **0.38** | **0.59** | **0.17** | **0.35** | **0.51** | **0.70** | **0.52** |
| | (8.7,0.15) | (9.0,0.15) | (6.2,0.13) | (3.5,0.10) | (5.0,0.13) | (12.2,0.35) | (4.0,0.09) | (4.8,0.08) |
| SRLD | **0.47** | **0.38** | **0.61** | **0.31** | **0.21** | **0.59** | **0.63** | **0.43** |
| | (8.4,0.10) | (9.1,0.10) | (6.1,0.10) | (3.2,0.06) | (5.6,0.11) | (11.1,0.23) | (4.4,0.08) | (5.2,0.06) |

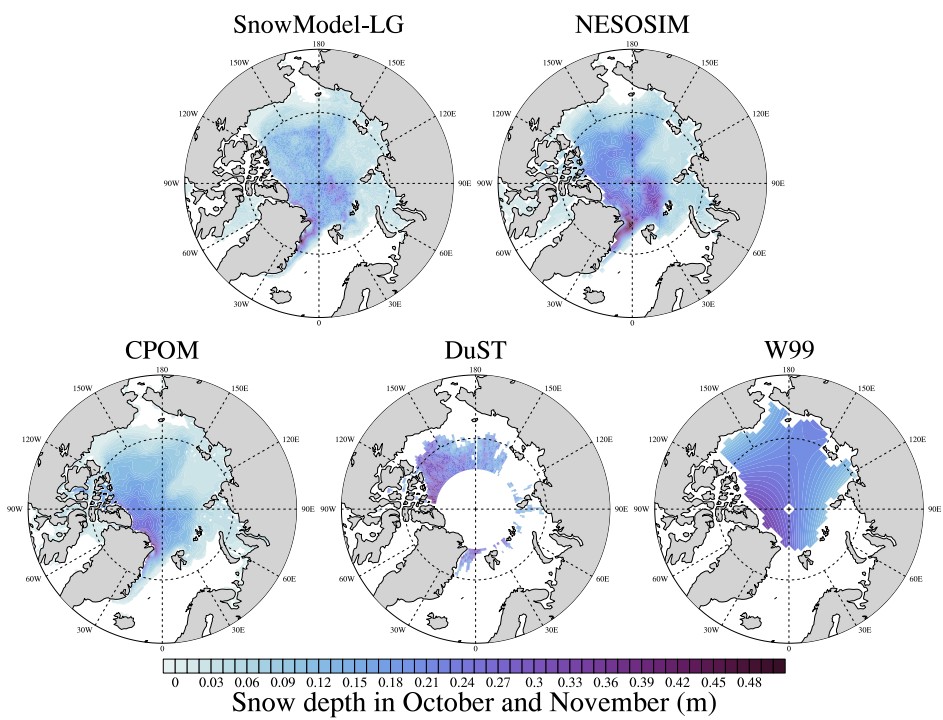

**Figure 1.** Mean snow depth (units: $m$) depth in autumn (October-November) 2014 for four snow products and $W99$.



**Figure 2.** Same as Figure 1 but in spring (March-April) among eight snow products, $W99$ and $SS18$.




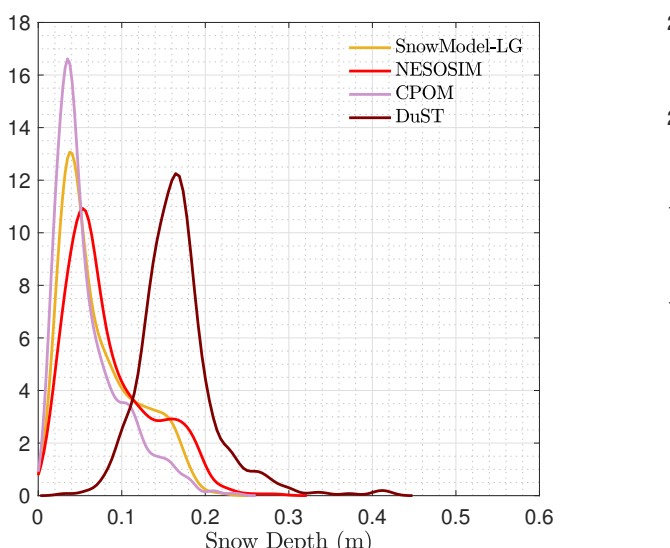
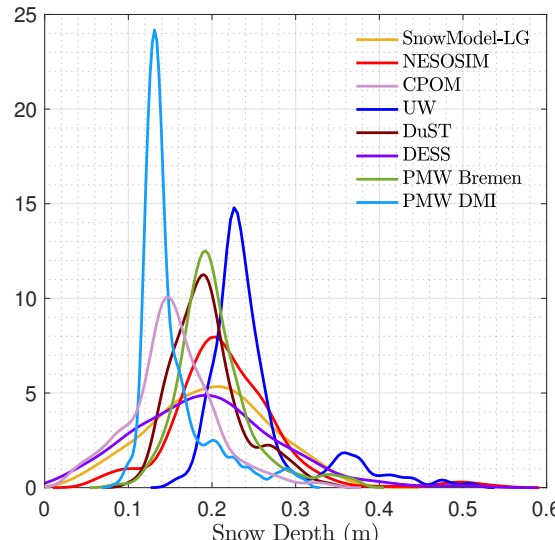

**Figure 3.** Snow distribution comparison within the common regions (in Figure S2(a)) in all snow products during the period 2000-2018 (different products cover different periods) in October-November (left: a) and March-April (right: b).

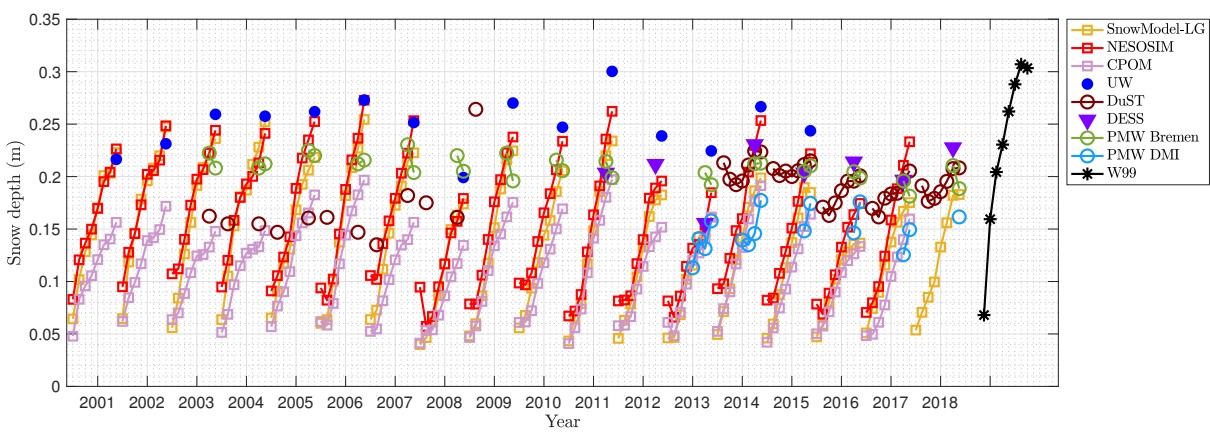

**Figure 4.** Time series of average monthly snow depth in each snow product within the common regions since 2000. Only winter time (September to next April) is shown.

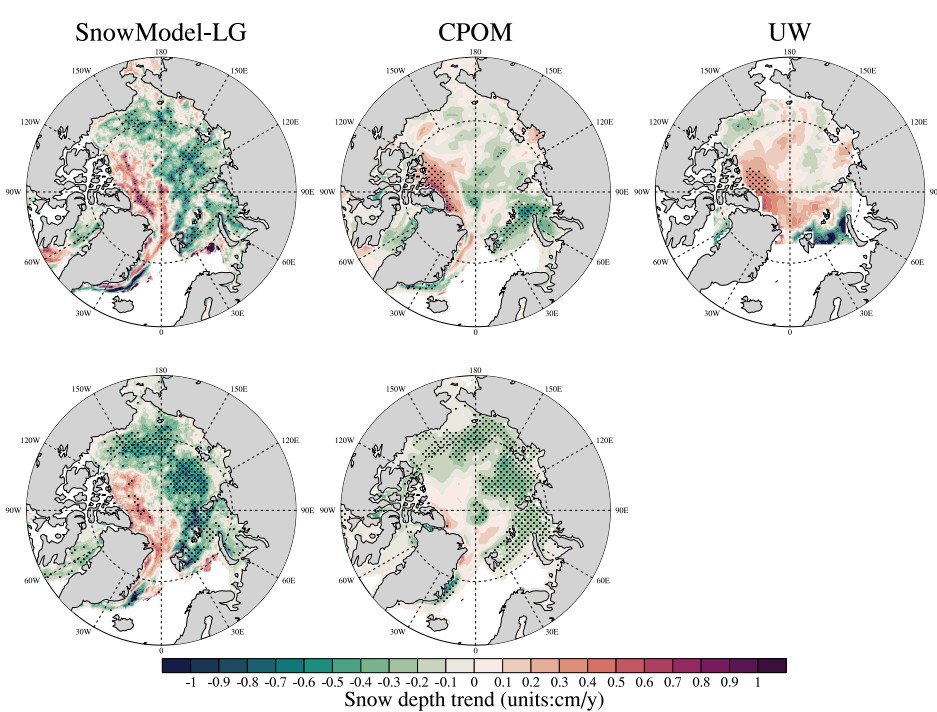

**Figure 5.** Trend of snow depth (Units: $cm/year$) for SnowModel-LG, CPOM and UW in spring (April) (first row) and autumn (November) (second row) during the period from 1991 to 2015. Areas with significant trends are shown as dotted areas (confidence level 95%).





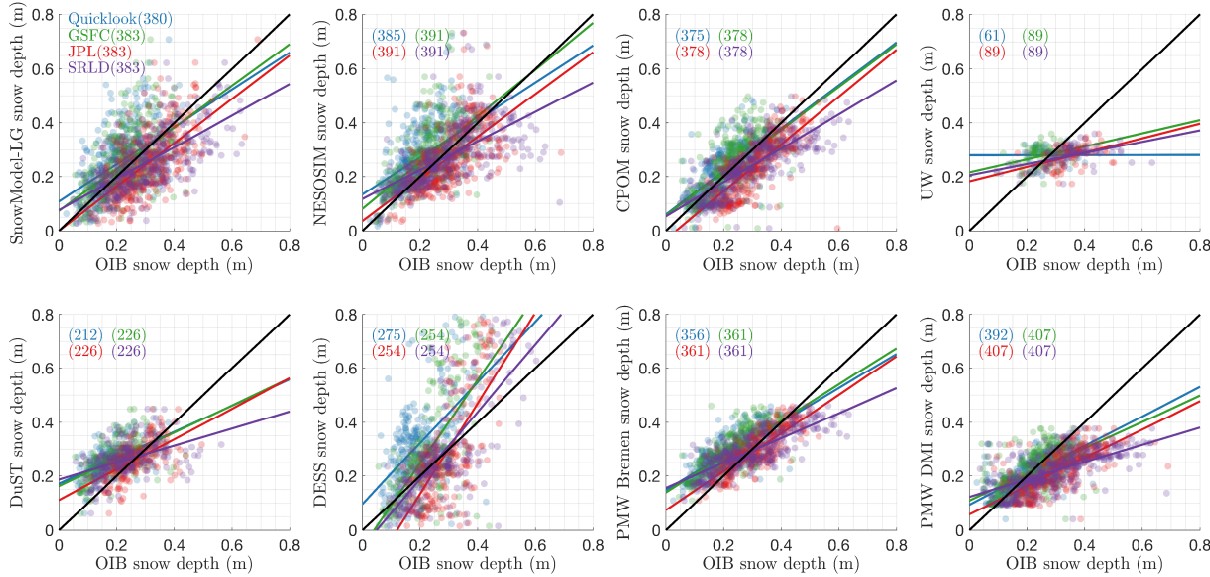

**Figure 6.** March and April comparison (only April in UW and March in DESS) between four monthly OIB products (quicklook in blue, GSFC in green, JPL in red and SRLD in purple) and all snow products in monthly onto $100km$ grid in the period of 2014 and 2015. Lines are best linear fits. Numbers in left corners indicate the valid count in the linear fitting.

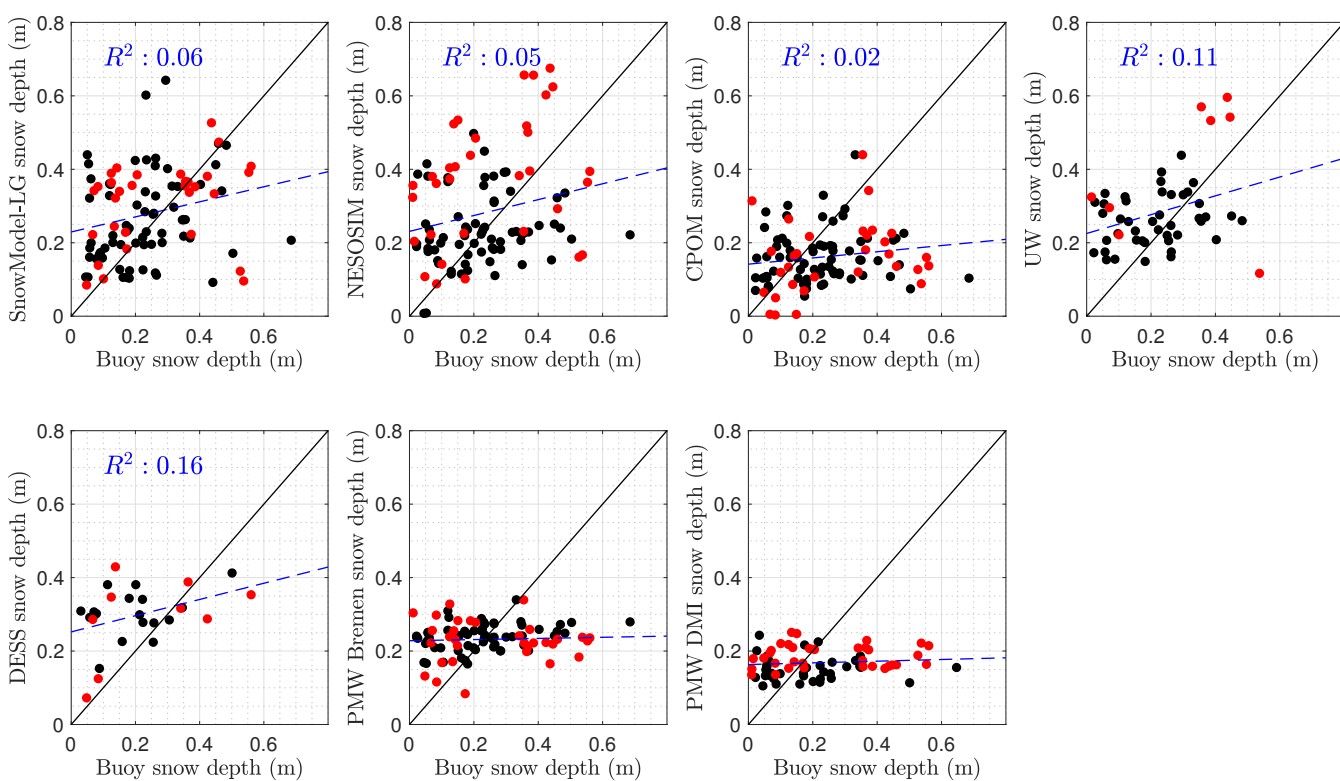

**Figure 7.** Comparison of average March/April snow depth in snow products in native spatial resolution versus the AWI (red dots)/IMBs (black dots) buoys. $R^2$ are given in blue.





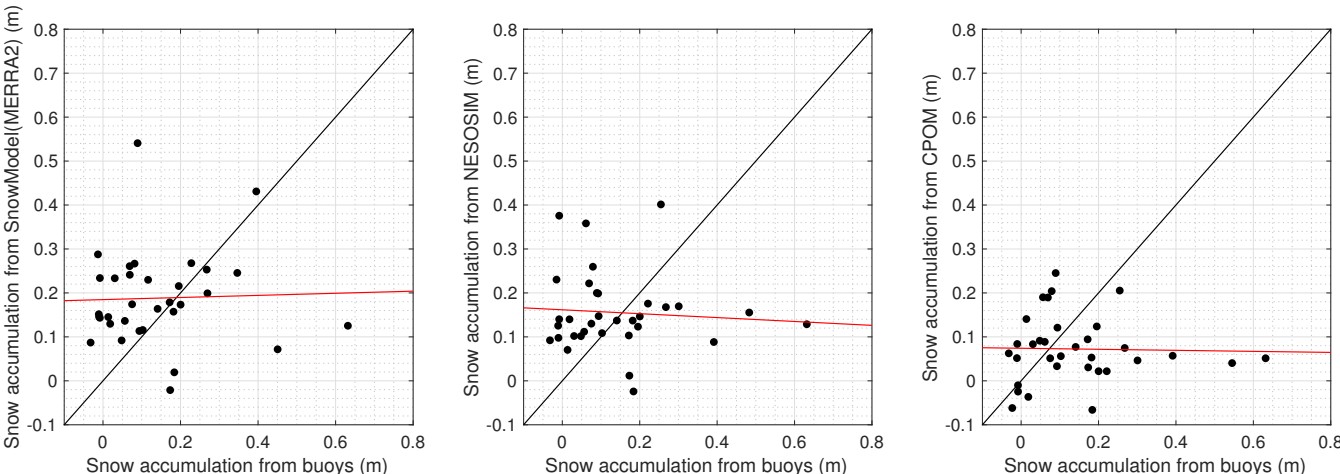

**Figure 8.** Comparison of total winter snow accumulation (starts from early winter to the next year) between buoys and three snow products based on daily snow products.

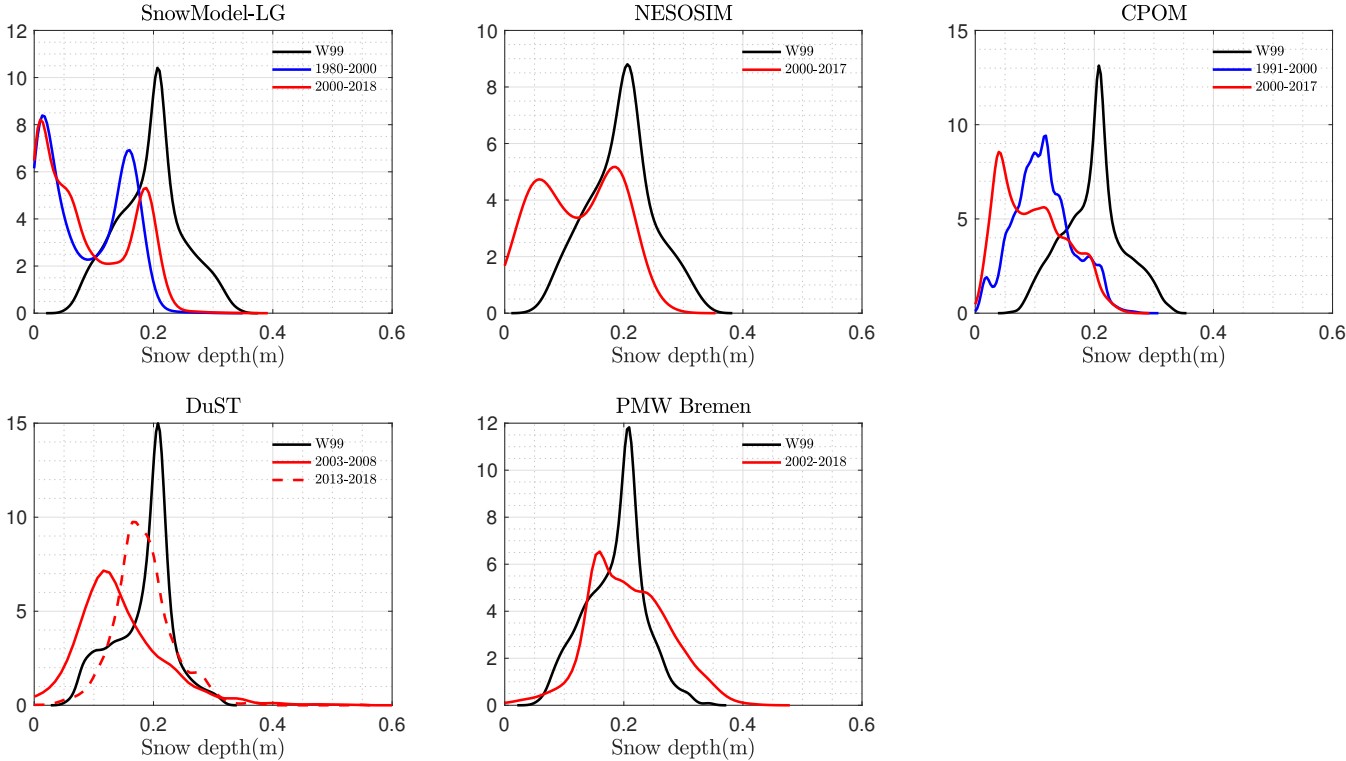

**Figure 9.** Snow distribution comparison between $W99$ and snow products in autumn (October/November).





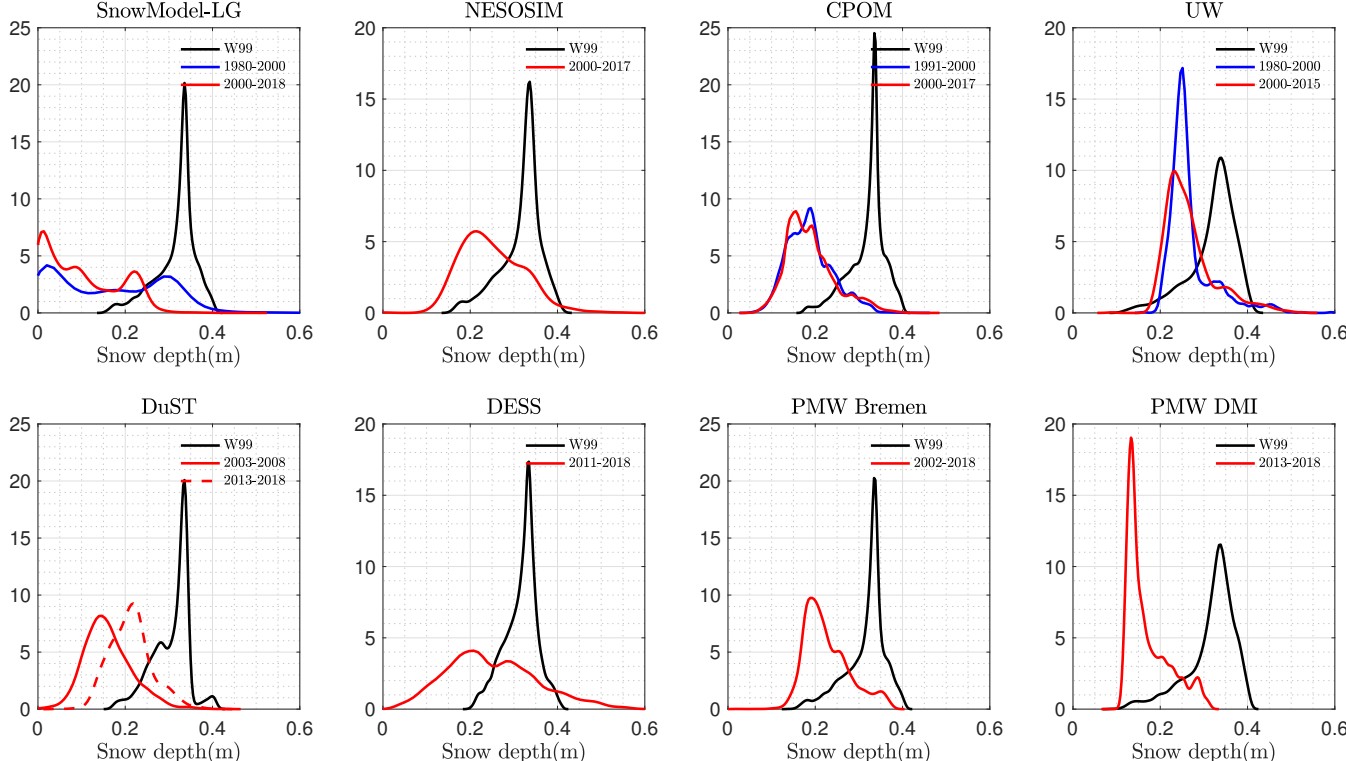

**Figure 10.** Snow distribution comparison between $W99$ and snow products in spring (March/April).





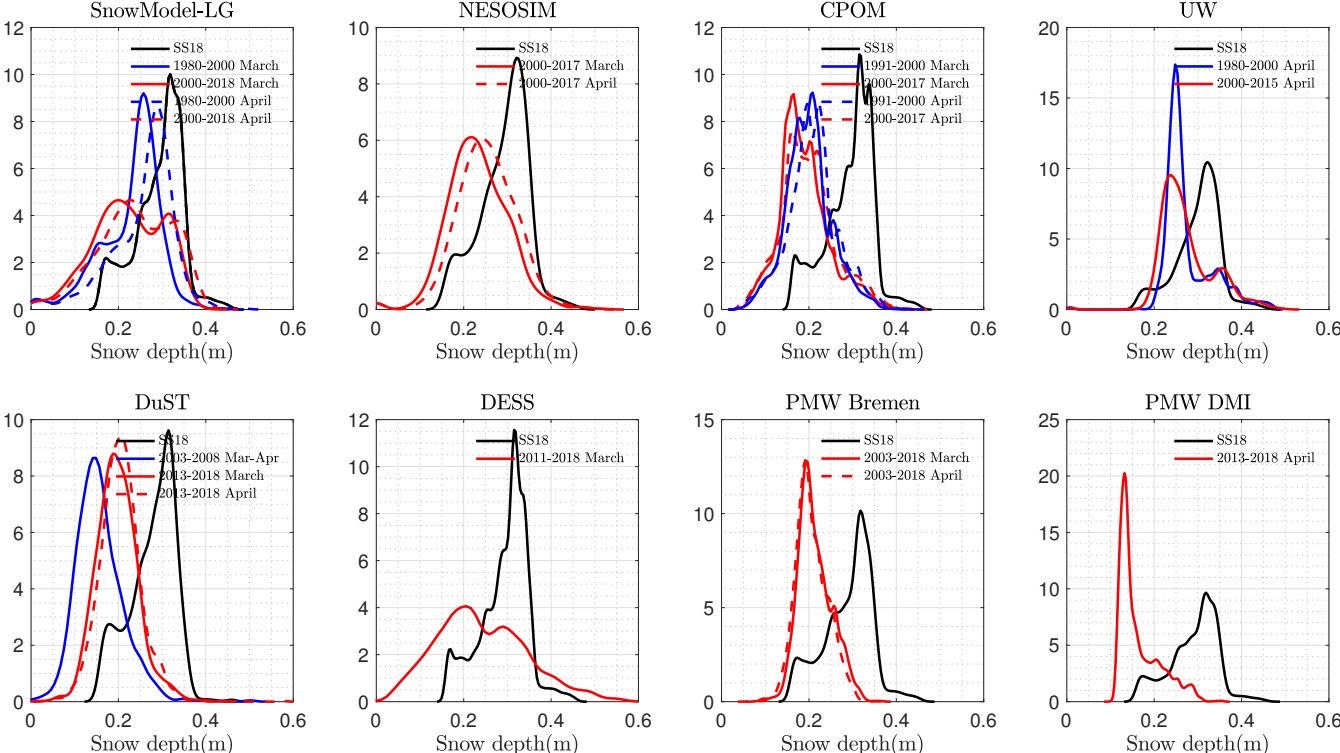

**Figure 11.** Snow distribution comparison between $SS18$ climatology and snow products.

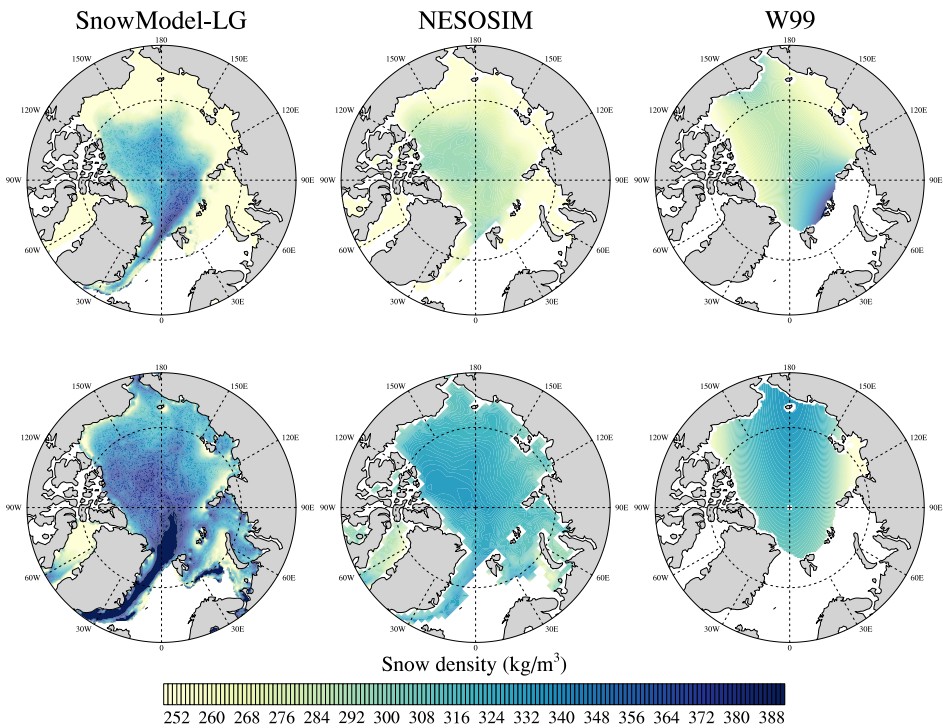

**Figure 12.** Mean snow density (Units: $kg/m^3$) according to SnowModel-LG, NESOSIM and W99 in November (first row) and the next April (second row) since 2000.





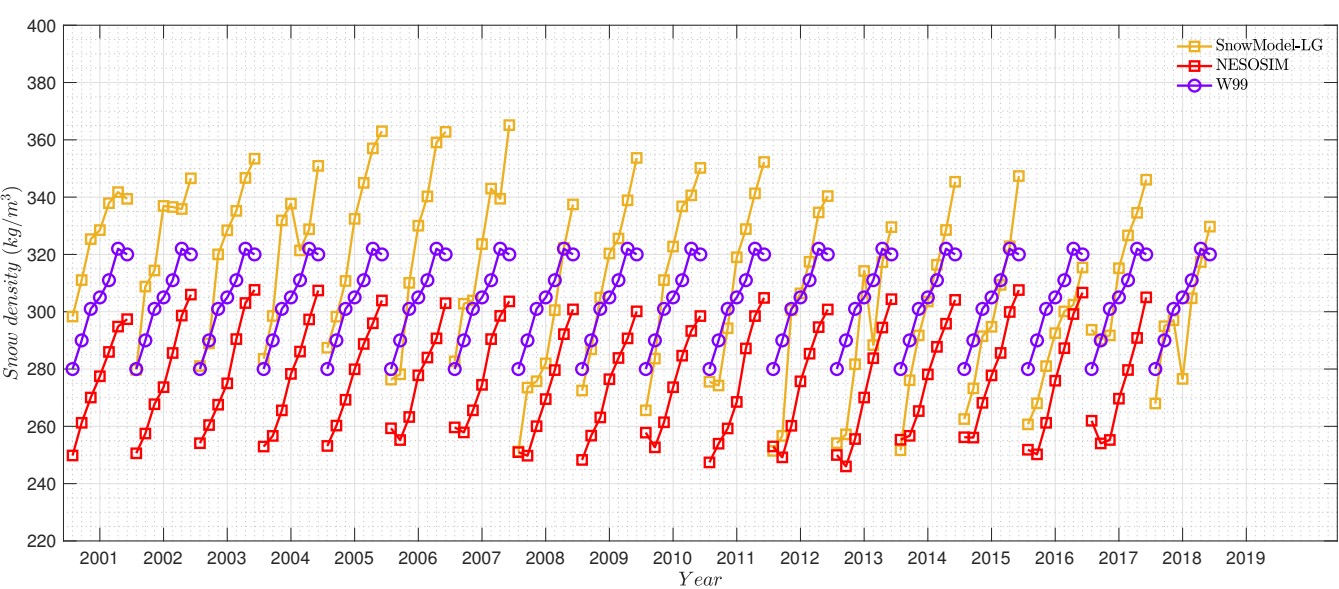

**Figure 13.** Time series of mean snow density (Units: $kg/m^3$) within Arctic basin (common region in Figure 1 $W99$) comparison in SnowModel-LG, NESOSIM and W99 since 2000.





**Figure 14.** Snow distribution of samples ((a), (b) and (c)) for original OIB measurements (black line), mean snow depth (dashed line), $1.6km$ segmentation re-sampling (strategy II: blue line) and $1.6km$ random re-sampling (strategy I: red line) within each cell. Typical fitting (d) between all OIB samples and a single sample under $1.6km$ segmentation, with fitting line in purple.