# Peer review of "Inter-comparison of snow depth over sea ice from multiple methods"

_The Cryosphere, 2020_

## Referee Comment (RC1) · Anonymous Referee #1 · 27 Apr 2020

Review of Inter-comparison of snow depth over sea ice from multiple methods by Zhou, Stroeve and others.

In this paper the authors explore how well various methods of determining the snow cover (mostly depth) across the sea ice of the Arctic Basin match each other, and where possible, match in-situ data. The methods consist of a) in-situ depth values from two ice buoy systems, b) Ice Bridge airborne measurements from radar, c) the Warren et al. (1999) and newer climatologies based on Russian measurements, d) satellite retrievals and e) physical models driven from reanalysis products. The authors find similarities between products in some geographic areas and at some periods of the winter, but also large discrepancies in both absolute values and patterns of depth. This comes as no surprise given the diversity of the ways in which the snow fields are

derived, and the limitations within the various methods.

This is a useful paper, and quite an impressive job in merely handling the massive data sets involved, but it does not go far enough in examining the "whys" of the discrepancies, and for that reason I think needs another round of revision. To simply state that radical differences in footprint sizes and spatial scales of the various snow depth products are at the root of the observed differences, while perhaps correct, seems of little practical use.

Part of the problem with the paper seems to start with this statement, given as a motivation for the study: (line 48) [to] provide an inter-comparison of these products so that recommendations can be made to the science community as to which data product best suits their needs.

I think I understand what the authors mean here, but it is not what they actually wrote. One would hope that the science community is not only looking for products that suit their needs, but in fact, are as accurate as possible. So while we all can accept that with footprint size and scaling issues, snow depth "truth" may be elusive, conceptionally it exists and it really is what users, and snow product developers, need to strive for. This murkiness in purpose reappears throughout the paper in that none of the models or methods are ever labeled as "wrong," even when the results seem to be utterly improbable. I understand no one wants to "slag" a model in print, but clearly one conclusion from the paper is that some of the models, in some situations, should be avoided for now (until improved) and the authors could be more explicit in saying so when that conclusion is clear (e.g. The PMW-DMI model has 13 cm as the end-of-winter average across the entire Basin: Fig. 3 left).

Before getting to the heart of my review, I wanted to raise a point that I am not an expert in but I believe is important. In any inter-comparison of models, the comparison will be skewed when the models are being forced by different reanalysis products, with the precipitation forcing, notoriously difficult to get right in the Arctic, varying a lot in both

time and space. How did the authors sort out input reanalysis precipitation differences from model biases. It seems to me that the models need to be run on the same input.

I found the results section of the paper quite good, and the graphics, of which perhaps there are a few too many, both useful and explanatory. If some shortening were to occur here, I would suggest the spatial trends in Autumn (Fig. 1), and the temporal trends over the period of record (Fig. 5) could be deleted. Autumn is a tough time to model or map snow, since open water (no platform) determines snow depth to a large extent, depending how it is handled in the model or with the satellite products. Figure 5, while interesting, is a dangerous figure to put out there, as it is likely to be cited as depicting real changes, when in fact the point of the paper is that the models and methods need significant improvement, as attested to in Figures 6, 7, 8, 9, 10 and 11. Personally, I would remove Figure 5. But if it is to be left in (see the next paragraph for this same point), then it would be more useful to explore why all of the models/methods converge in getting increasing snow depth in the Greenland-Canadian sector, while the Russian-Siberia sector is losing snow then to just put it out there as "trends." Can this convergence be traced back to some basic climate variables that enters all of the various models? What drives the convergence?

And that type of analysis is what I think the paper currently lacks. For example, starting with Figure 3, depth histograms offer a wealth of interpretive information if examined closely. SnowModel and NESOSIM show a distinct drift shoulder (deeper snow) in Autumn that indicates they allow the snow to drift immediately, while CPOM and DuST show little or none. So this feature is about drifting: is it real? Should it be there? Strangely, the drift shoulder disappears by Spring for SnowModel and NESOSIM, yet appears in UW and DuST. Therein lies some behavioral differences that could provide insights for model/method improvement.

Figure 4 offers similar analysis possibilities. A lot is known at least at local scales about the patterns of snow build up on sea ice in various locations. The slope of these seasonal trajectories, and whether they curve off late in the season or not, is

a diagnostic that could readily provide insight into what is or isn't working. The W99 curve is suggestive; only the CPOM curves seem to carry that shape, yet these curves fail to reach reasonable depth values by Spring. That should tell us something. Finally, considering Figure 10, which is fascinating, the paragraph (lines 426-431) discussing it barely scratches what could be gleaned from the data. Not only is W99 significantly deeper than the model/method products, in some cases by more than 2X, but also the histogram shapes are so different as to look like they are from totally different fields. SnowModel, DuST, and DESS all have a zero snow fraction, while the others do not. PMW-DMI is as peaked as the W99 data, but is about 1/3rd as deep. Surely, contained in this plot, which took considerable effort to develop, are many useful suggestions for model and method improvement (most of the same comments apply to Fig. 11 and SS18), but to get to them requires thinking about why the histograms have the shape they do. In the old days, a lot of emphasis was place on interpreting skewness and kurtosis, and that literature might be of use here.

In summary, I recommend this manuscript be returned to the authors for revisions without acceptance, that they be commended for undertaking a very useful and difficult set of analyses, and that they be urged now to reap the rewards of that analyses by gleaning more pertinent and useful information from their work. That that could prove very useful in improving existing and future models and methods of extrapolating snow over the Arctic Basin.

---

## Referee Comment (RC2) · Anonymous Referee #2 · 2 Jul 2020

This paper is reporting inter-comparison of various snow products from all different sources. The topic is timely and important – an overall good attempt. However, the manuscript suffers from a bad description of the results, especially Section 3. I recommend revising the whole section to point each argument to relevant evidence (figure or table) to support, also the supplementary materials should be there to support the main results, so suggest avoiding unnecessary explanation of the supplementary materials (that can be in the caption). In the beginning, I was quite excited to read the manuscript but quickly losing the interests due to the way the results being described. Section 3 can be much improved (I think it is Results and Discussions) if the authors restructure and rewrite them carefully. Too much unnecessary description in my view, and too little discussions. I really value the topic, but unfortunately, the manuscript itself is not up to

the standard.

Specific comments: Section 3 – Each sentence should be backed with Figures or Tables for the evidence. Many sentences are without pointing to specific figures. See some examples below.

Line 284: In particular, "NEOSIM. . .section" Should point the readers to figures or tables for your argument.

Line 293: "Deeper. . .-LG." "DESS shows. . ." Are you referring to Figure S1 or others? I recommend revising all sentences for this.

Line 301: "Modal and distribution. . ." do you need this? It is obvious from the figure.

Line 304: "Since the. . ." then what are the spatial coverage for other products in comparison?

Line 305: Are you referring Figure 3b or else?

Line 306: "PMW. . ." where do the readers to look at? "Mean snow depth. . .2.0 cm higher. . ." Is it spring or autumn? Please be specific and add figures.

Line 313: "We additionally. . .set used." The whole paragraph is describing a supplementary figure. What is the key point for this? I see a general tendency that the description of the results is heavily on supplementary materials. The supplementary materials are to support the main figures and tables but seems overpowering. For this paragraph, please think whether you need all detailed description of the figure, rather than using them to support the main results. I found this is the problem throughout the manuscript.

Line 324: "DESS exhibits. . .snow depths." Which figure or table for this?

Line 325: Are we still in Figure 4?

Line 325: winter time snow accumulation is largest in SnowModel-LG. . .Are you refer-

ring to Figure 4? I don't see it clearly.

Line 331: "the inter-annual variability of monthly averaged..is small among" I don't get this. Small among all snow products? Small compared to what? What do you mean? Also it is difficult to see which points are November or April in Figure 4.

Line 335: "DuST show a significant positive..." where is the evidence?

Line 337: "This features is also not..." where is the evidence?

Line 339: "We do not find..." Where can I see that? Figure 5 shows the trend from 1991 to 2015.

Line 353: "...directly fitted against OIB". What do you mean? You mean OIB data assimilated into those products?? "...show high correlations..." where is the number?

Line 357: "Figure 6..." Shouldn't this be first mentioned? Or Do you need this sentence? "The corresponding..." If you directly reference in the text, why do you need this?

Line 362: "Not surprisingly..." Why?

Line 380: What is the key message here? Higher R2 for coarse resolution but no significant difference for temperature resolution?

Line 387: "Given..." it seems the results are sensitive to the choice of OIB data, but to me it does show some consistency, e.g., SnowModel-LG R2 range from 0.27 to 0.47 yet PMW Bremen from 0.56 to 0.70. So, I don't know whether I should agree that it is impossible to conclude which ones perform better. Perhaps it is more related to how the particular products dependent to OIB data in their production?

Line 397: "PMW Bremen and..." no variability where I can see that? Figure 7? You mean no correlation? You have Figure 7 as one of the main figure but very little description for that. What is the point showing this?

Line 400: I think it is far-fetched to compare each buoy with such products. I wonder why we see no correlation. Section 3.4: Personally, I like the results from this section, and found interesting among other results. The difficulty is that it is like comparing apple with an orange, but it does give us a general conclusion which is useful.

5. Discussion: I found this is confusing about the scope of this research. Need to make it clear whether you want to show the inter-comparison results or developing a new sampling strategy. The whole discussions are about sampling strategy not about inter-comparison results. This actually tells me whether this is a research article or just for a discussion.

6. Conclusions: It is too lengthy. Should be cut down to the key results and messages.

---

## Author Comment (AC1) · 20 Aug 2020

The authors would like to thank the referee for the prompt and precise comments to our reply. We want to re-emphasize that the purpose of the study is to investigate the current status-quo of the community in snow cover reconstruction and retrieval for Arctic sea ice. The variety of the methods used in the products, as well as the potential lack of independent data pose unique challenges, but we consider this a timely update of the community's efforts, and the collateral trade-offs during the analysis necessary at this stage of research progress. Moreover, the general consistency among many of the products reflect the exciting progress that has been made, and provide confidence for further improvements of snow depth retrieval. Also, messages are conveyed relating to both observation and representation issues. In specific, with respect to the comments from the referee, we have made replies and accompanying revisions as follows.

***The referee's comments 1:***

*Review of Inter-comparison of snow depth over sea ice from multiple methods by Zhou, Stroeve and others.*

*In this paper the authors explore how well various methods of determining the snow cover (mostly depth) across the sea ice of the Arctic Basin match each other, and where possible, match in-situ data. The methods consist of a) in-situ depth values from two ice buoy systems, b) IceBridge airborne measurements from radar, c) the Warren et al. (1999) and newer climatologies based on Russian measurements, d) satellite retrievals and e) physical models driven from reanalysis products. The authors find similarities between products in some geographic areas and at some periods of the winter, but also large discrepancies in both absolute values and patterns of depth. This comes as no surprise given the diversity of the ways in which the snow fields are derived, and the limitations within the various methods.*

*This is a useful paper, and quite an impressive job in merely handling the massive data sets involved, but it does not go far enough in examining the "whys" of the discrepancies, and for that reason I think needs another round of revision. To simply state that radical differences in footprint sizes and spatial scales of the various snow depth products are at the root of the observed differences, while perhaps correct, seems of little practical use.*

*Part of the problem with the paper seems to start with this statement, given as a motivation for the study: (line 48) [to] provide an inter-comparison of these products so that recommendations can be made to the science community as to which data product best suits their needs. I think I understand what the authors mean here, but it is not what they actually wrote. One would hope that the science community is not only looking for products that suit their needs, but in fact, are as accurate as possible. So while we all can accept that with footprint size and scaling issues, snow depth "truth" may be elusive, conceptionally it exists and it really is what users, and snow product developers, need to strive for. This murkiness in purpose reappears throughout the paper in that none of the models or methods are ever labeled as "wrong," even when the results seem to be utterly improbable. I understand no one wants to "slag" a model in print, but clearly one conclusion from the paper is that some of the models, in some situations, should be avoided for now (until improved) and the authors could be more explicit in saying so when that conclusion is clear (e.g. The PMW-DMI model has 13 cm as the end-of winter average across the entire Basin: Fig. 3 left).*

**Reply:**

The authors understand the urge for the 'accurate' snow product for the community. The motivation of the paper is to investigate the status and (to try) to attribute the discrepancies among current snow products constructed from multiple

methods, and further validate against available observations. However, as discussed in Section 4.3, due to the resolution differences of the various snow depth measurements, there exist distinctive representation issues when local snow depth measurements are used. Therefore, it is hard to judge which is the "true" snow depth based on limited validation datasets.

Although it is impossible to pick one "perfect" snow product, we do find outliers. For example, the positive trend in DuST snow depths during ICESat and CryoSat-2 periods is caused by the limitation in processing the calibration with OIB in the product, which is inconsistent with other products. In spite of lacking thorough analysis of independency and representative issues, we pointed out "unreasonable" performances in some products, including low correlation with OIB in UW product and no correlation and small snow variability in the PMW Bremen and PMW DMI product and buoy comparison although existing representation issues. The above four products are therefore not recommended.

According to the comments, he revised paper further emphasizes the choices of the candidate products with consistent performance according to our analysis. Regarding the manuscript, relevant revisions are highlighted in yellow (Line 311; Line 337; Line 361; Line 388; Line 542).

**_The referee's comment 2:_**

_Before getting to the heart of my review, I wanted to raise a point that I am not an expert in but I believe is important. In any inter-comparison of models, the comparison will be skewed when the models are being forced by different reanalysis products, with the precipitation forcing, notoriously difficult to get right in the Arctic, varying a lot in both time and space. How did the authors sort out input reanalysis precipitation differences from model biases. It seems to me that the models need to be run on the same input._

**Reply:**

The authors are working with data products as provided by various contributors. For the four reanalysis-based products, contributors use different reanalyses as input. Furthermore, the methodologies differ a lot as well. Regarding reanalyses, a comprehensive assessment of available products was made recently by Barrett et al. (2020). They found inter annual variability in reanalysis precipitation was consistent among the different products, but that MERRA-2 was wetter than ERA-I or ERA-5. Spatial patterns are however consistent. Further, Stroeve et al. (2020) also found that running SnowModel-LG with ERA-5 and MERRA-2 gave snow depths within a few cm of each other. We anticipate that the differences between the various reanalysis-based estimates has more to do with how snow processes related to sea ice are considered, such as the snow accumulation or loss and physical process, rather than the reanalysis product used. The distinctive snow initial conditions between SnowModel-LG and NESOSIM is another example of hurdles in aligning these products. More aligned comparison of the reanalysis-based products, which is suggested by the referee, is beyond the scope of this study, including attributing to uncertainties (or differences) to driving datasets and methods. This would potentially require another round of intercomparison with much higher coordinated activities across the contributing institutes.

*The referee's comment 3:*

*I found the results section of the paper quite good, and the graphics, of which perhaps there are a few too many, both useful and explanatory. If some shortening were to occur here, I would suggest the spatial trends in Autumn (Fig. 1), and the temporal trends over the period of record (Fig. 5) could be deleted. Autumn is a tough time to model or map snow, since open water (no platform) determines snow depth to a large extent, depending how it is handled in the model or with the satellite products. Figure 5, while interesting, is a dangerous figure to put out there, as it is likely to be cited as depicting real changes, when in fact the point of the paper is that the models and methods need significant improvement, as attested to in Figures 6, 7, 8, 9, 10 and 11. Personally, I would remove Figure 5. But if it is to be left in (see the next paragraph for this same point), then it would be more useful to explore why all of the models/methods converge in getting increasing snow depth in the Greenland-Canadian sector, while the Russian-Siberia sector is losing snow then to just put it out there as "trends." Can this convergence be traced back to some basic climate variables that enters all of the various models? What drives the convergence?*

**Reply:**

Thank you for referee's suggestion. Figure 1 is removed from the manuscript, and moved into the supplementary material as Figure S1. Further explanation is added about newly Figure 7 (denoted as Figure 5 in old version). As mentioned in the paper, snow depth decreases over Eurasia as a result of delayed freeze-up while the increasing over Greenland-Canadian sector from the three reanalysis-based products (SnowModel-LG, NESOSIM and CPOM) use precipitation (or snowfall) from reanalysis. Existing studies of reanalysis-based precipitation in the Arctic indicate more snowfall and ensuing accumulation, hence thicker snow. Serreze et al. (2012) found that in summer and early autumn, the precipitable water from MERRA, CFSR and ERA-Interim both show positive trend in the Canadian Arctic Archipelago, and Stroeve et al. (2020) suggests that widely open water may increase winter precipitation and further affect snow accumulation. Further, the snow accumulation process largely depends on ice motion fields and the above all three products (SnowModel-LG, NESOSIM and UW) use NSIDC ice drift, which may potentially cause similar pattern in snow depth trend.

*The referee's comment 4:*

*And that type of analysis is what I think the paper currently lacks. For example, starting with Figure 3, depth histograms offer a wealth of interpretive information if examined closely. SnowModel and NESOSIM show a distinct drift shoulder (deeper snow) in Autumn that indicates they allow the snow to drift immediately, while CPOM and DuST show little or none. So this feature is about drifting: is it real? Should it be there? Strangely, the drift shoulder disappears by Spring for SnowModel and NESOSIM, yet appears in UW and DuST. Therein lies some behavioral differences that could provide insights for model/method improvement.*

**Reply:**

In general, the authors want to explain that the observable modes in the PDF is caused by snow accumulation on different ice types. The shoulder shape in Figure 2 mentioned by the referee is the second mode in PDF. This bimodal PDF is more obvious in Figure S3 both for SnowModel-LG, NESOSIM and CPOM as a result of different snow accumulation over FYI and MYI. MYI may accumulates initial snow

cover early in autumn, while FYI won't be able to catch any snow until it forms. Another minor issue in the comparison is that, the shoulder shape in Figure 2.a is due to the smaller common coverage area compared with Figure S3. The common coverage without DuST (shown in Figure S2) covers more over Canadian coastal regions, where MYI manifests. All spring snow PDFs in Figure S3.b show the bimodal or long-tail features, which is also found in other observations (Kwok et al., 2011).

***The referee's comment 5:***

*Figure 4 offers similar analysis possibilities. A lot is known at least at local scales about the patterns of snow build up on sea ice in various locations. The slope of these seasonal trajectories, and whether they curve off late in the season or not, is a diagnostic that could readily provide insight into what is or isn't working. The W99 curve is suggestive; only the CPOM curves seem to carry that shape, yet these curves fail to reach reasonable depth values by Spring. That should tell us something. Finally, considering Figure 10, which is fascinating, the paragraph (lines 426-431) discussing it barely scratches what could be gleaned from the data. Not only is W99 significantly deeper than the model/method products, in some cases by more than 2X, but also the histogram shapes are so different as to look like they are from totally different fields. SnowModel, DuST, and DESS all have a zero snow fraction, while the others do not. PMW-DMI is as peaked as the W99 data, but is about 1/3rd as deep. Surely, contained in this plot, which took considerable effort to develop, are many useful suggestions for model and method improvement (most of the same comments apply to Fig. 11 and SS18), but to get to them requires thinking about why the histograms have the shape they do. In the old days, a lot of emphasis was place on interpreting skewness and kurtosis, and that literature might be of use here.*

**Reply:**

The authors have included more analysis of the seasonal cycle in the manuscript (now a dedicated section, Sec. 3.2). Regarding to the comment on the specific details of intercomparison, the authors agree that the seasonal curve shape in CPOM is similar with that in W99, however, SnowModel-LG, NESOSIM also has the similar curve shapes in some years, especially from 2011 to 2012 and from 2015 to 2016. However, the interannual variability of this seasonal shape varies widely in all reanalysis-based products. As sea ice freeze-up has happened later and later over the last 40 years, it is expected that the snow accumulation during the early stage of winter is no longer similar to W99.

As in Figure 3 and S5 (which are Figure 8 and 9 in the previous version of the manuscript), the authors apologize that the previous result of the comparison between SnowModel-LG and W99 was not confined into Arctic basin, which is not the same region as the other products. These two plots are updated into the paper and the discussion part is revised. Figure 3 basically shows the results of Figure 2 but for different regions. In Figure 2, common regions only include Baffin Bay, north of Barents and Kara Sea but in Figure 3 and S5, only the Arctic basin is included (as shown by W99 in Figure 1). Within the Arctic basin, thin snow (less than 10cm) is observed in SnowModel-LG, DuST and DESS, while NESOSIM, CPOM, UW, PMW Bremen and PMW DMI show deeper snow packs. Other studies have indicated that snow depth is decreasing over the last 30 years (Webster et al., 2014), and that the

snow depth since 2000s is thinner than in W99 and SS18. The majority of the snow products are consistent with this decreasing snow depth, but their mean values and slopes differ.

As measurements in W99 are mainly over the western of Arctic, where more of the ice was MYI, SS18 is adopted for providing more details over eastern Arctic, where FYI dominates. The shapes of W99 and SS18 PDF are different as a result of spatial coverage and the inclusion of FYI. Since the ice is getting younger, the histogram shapes of the various snow products are expected to differ from W99 and SS18 especially after 2000. Figure 3-4 also provide snow depth estimations during 1980s or 1990s from reanalysis-based products, suggesting how snow depth has changed in each product over the longer time series. The mode in SnowModel-LG is decreasing while that in UW is still unchanged over the 40 years. The authors agree that the comparison with climatology is helpful for model improvements and we have revised relevant parts, highlighted in red (Line 320; Line 325). As for the skewness and kurtosis of snow depth distribution, Kwok et al, (2011) found that the snow depth is left skewed from OIB and snow depth from ICESat-2 and CryoSat-2 (Kwok et al., 2020) also shows the left skewness especially during early winter. PDF shapes from the snow products and OIB observations is quite different from W99 and SS18.

***Summary comments:***

*In summary, I recommend this manuscript be returned to the authors for revisions without acceptance, that they be commended for undertaking a very useful and difficult set of analyses, and that they be urged now to reap the rewards of that analyses by gleaning more pertinent and useful information from their work. That that could prove very useful in improving existing and future models and methods of extrapolating snow over the Arctic Basin.*

**Reply:**

The authors sincerely thank the referee for the comments. Revisions to the manuscript have been made to: (1) restructure Section 3 to 5 for better clarity, with each subsection covering an aspect of intercomparison, and (2) include more analyses of the intercomparison results, especially for the consistency of the products and PDF of snow depth and comparison with climatology, and (3) explicitly picking out the consistent products and outliners. We sincerely hope that through these revisions, we can convey more clear and informative results.

**Reference:**

Barrett, A. P., Stroeve, J., and Serreze, M. C.: Arctic Ocean Precipitation from Atmospheric Reanalyses and Comparisons with North Pole Drifting Station Records, Journal of Geophysical Research: Oceans, 2020.

Kwok, R., et al. Airborne surveys of snow depth over Arctic sea ice. Journal of Geophysical Research: Oceans 116.C11, 2011.

Kwok, R., Kacimi, S., Webster, M. A., Kurtz, N. T., & Petty, A. A. Arctic Snow Depth and Sea Ice Thickness From ICESat-2 and CryoSat-2 Freeboards: A First Examination. Journal of Geophysical Research: Oceans, 125(3), e2019JC016008, 2020.

Serreze, M. C., Barrett, A. P., and Stroeve, J.: Recent changes in tropospheric water vapor over the Arctic as assessed from radiosondes and atmospheric reanalyses, Journal of Geophysical Research: Atmospheres, 117, 2012.

Stroeve, J., Liston, G. E., Buzzard, S., Zhou, L., Mallett, R., Barrett, A., Tschudi, M., M. Tsamados, P. I., and Stewart, J. S.: A Lagrangian snow-evolution system for sea-ice applications (SnowModel-LG): Part II - Analyses., Journal of Geophysical Research: Oceans, in revision, 2019.

Webster, M. A., Rigor, I. G., Nghiem, S. V., Kurtz, N. T., Farrell, S. L., Perovich, D. K., and Sturm, M.: Interdecadal changes in snow depth on Arctic sea ice, Journal of Geophysical Research: Oceans, 119, 5395–5406, 2014.

---

## Author Comment (AC2) · 20 Aug 2020

The authors would like to thank the referee for the prompt and precise comments to our reply. We have thoroughly revised the manuscript to include the presentation of the results, as well as the analysis of the intercomparison results. As follows, we have made replies and the accompanying revisions.

*The referee's comments 1:*

*This paper is reporting inter-comparison of various snow products from all different sources. The topic is timely and important – an overall good attempt. However, the manuscript suffers from a bad description of the results, especially Section 3. I recommend revising the whole section to point each argument to relevant evidence (figure or table) to support, also the supplementary materials should be there to support the main results, so suggest avoiding unnecessary explanation of the supplementary materials (that can be in the caption). In the beginning, I was quite excited to read the manuscript but quickly losing the interests due to the way the results being described. Section 3 can be much improved (I think it is Results and Discussions) if the authors restructure and rewrite them carefully. Too much unnecessary description in my view, and too little discussions. I really value the topic, but unfortunately, the manuscript itself is not up to the standard.*

*Specific comments: Section 3 – Each sentence should be backed with Figures or Tables for the evidence. Many sentences are without pointing to specific figures. See some examples below. Line 284: In particular, "NEOSIM...section" Should point the readers to figures or tables for your argument.*

**Reply:**

A great thanks for the referee's suggestion! Accordingly, we have reformulated the contents of Sect. 3 through Sect.5 to improve the structure, and rewritten the majority of the contents. In the revised manuscript, Sect.3 includes the major results of intercomparison: Sect.3.1 covers basic climatology, Sect.3.2 the seasonal cycle, and Sect.3.3 the long-term trend and inter-annual variability. Sect.4 now includes the validation results: Sect.4.1 covers study with OIB, Sect.4.2 that with buoys, and Sect.4.3 the discussion of representation issues.

Regarding the specific comment, the authors have changed the Line 284 into 'In particular, Figure 1 indicates that the thickest snow in late winter/early spring for NESOSIM manifests in the East Greenland Sea, while in DESS, the deepest snow is concentrated in the Canadian Arctic.'.

*The referee's comments 2:*

*Line 293: "Deeper. . .-LG." "DESS shows. . ." Are you referring to Figure S1 or others? I recommend revising all sentences for this.*

**Reply:**

The authors have revised the whole paragraph to read as follows 'During autumn, for the region north of Greenland and Svalbard, SnowModel-LG runs forced with MERRA-2 show a similar spatial pattern as other reanalysis based modeling systems (i.e. NESOSIM and CPOM), but shallower snow than NESOSIM and slightly deeper snow (Figure S2). DuST also shows the thickest snow pack in this region (16.0 cm mean snow depth), though the spatial coverage is more limited. Spring snow depth, ranging from 25.0cm to 30.0cm in the Arctic domain (Figure S1), exhibits large spatial variability among all products. Relatively thicker snow packs in the North Atlantic sector are evident in all reanalysis-based products except for CPOM. Deeper snow packs are expected in this region as it receives precipitation from the North

Atlantic storm tracks. For comparison, snow is also the deepest (over 35.0cm) to the north of Svalbard in both the W99 and SS18 climatologies. NESOSIM further suggests thick snow over Davis Strait, with spring averaged snow depths greater than 25.0cm. This is in stark contrast to the other data sets over the FYI in that region, and is likely unrealistic given this is a region of first-year ice that does not usually freeze until December/January (e.g. Stroeve et al. 2014), limiting the time over which snow can accumulate on the ice.'

***The referee's comments 3:***

*Line 301: "Modal and distribution. . ." do you need this? It is obvious from the figure.*

**Reply:**

The authors have deleted the sentence in Line 301 and the sentence is rewritten as 'Out of all the reanalysis-based data products, snow depth distributions in NESOSIM are shifted towards slightly deeper snow packs (8.0cm) than those from SnowModel-LG (7.0cm) and CPOM (6.0cm) during autumn, although the shapes of the distributions are similar.'

***The referee's comments 4:***

*Line 304: "Since the. . ." then what are the spatial coverage for other products in comparison?*

**Reply:**

These histogram comparisons are limited to regions below 81.5°N, which is emphasized in Line 298. That means that the histograms of all products are also constrained to the same region (Figure S1.a).

***The referee's comments 5:***

*Line 305: Are you referring Figure 3b or else?*

*Line 306: "PMW..." where do the readers to look at? "Mean snow depth...2.0 cm higher. . ." Is it spring or autumn? Please be specific and add figures.*

**Reply:**

The whole paragraph describes the histogram comparison in Figure 2. The part is rewritten to highlight the key points and remove the unnecessary information.

***The referee's comments 6:***

*Line 313: "We additionally. . .set used." The whole paragraph is describing a supplementary figure. What is the key point for this? I see a general tendency that the description of the results is heavily on supplementary materials. The supplementary materials are to support the main figures and tables but seems overpowering. For this paragraph, please think whether you need all detailed description of the figure, rather than using them to support the main results. I found this is the problem throughout the manuscript.*

**Reply:**

The authors have rearranged the content of the results part and changed the whole paragraph as: 'Additionally, we examine snow over the three different sectors in spring 2015 (Figure S4). The thickest snow from reanalysis-based snow products is mainly over the North Atlantic while satellite-based products indicate more snow accumulating over CA. Although this is only one year of comparison, it shows that

regional differences in snow accumulation can be quite pronounced depending on data set used.'

**The referee's comments 7:**

Line 324: "DESS exhibits. . .snow depths." Which figure or table for this?

**Reply:**

The authors add Sec. 3.3 to discuss the interannual variability and trends in these products. Figure 5 mainly provides detailed knowledge on interannual variability. Specifically, the interannual variability consistency analysis about DESS and other reanalysis-based products is for March from 2011 to 2018.

**The referee's comments 8:**

Line 325: Are we still in Figure 4?

**Reply:**

The snow accumulation is discussed in Section 3.2 and newly added Figure 6 further helps the analysis of seasonal cycle of different products. For example, the paper finds that the snow in W99 accumulates more during early winter and thus the seasonal curves are flattened near the end of winter. However, SnowModel-LG, NESOSIM and CPOM share a similar seasonal accumulation curve but different from W99 during late winter.

**The referee's comments 9:**

Line 325: winter time snow accumulation is largest in SnowModel-LG. . .Are you refer ring to Figure 4? I don't see it clearly.

**Reply:**

The newly added Figure 6 clearly shows the distinct accumulations in SnowModel-LG, NESOSIM and CPOM. And the paper highlights that the overall difference by the end of winter between the SnowModel-LG and NESOSIM are less pronounced, with respect to their autumn conditions. This may be partially due to the fact that the initial condition for wintertime accumulation in NESOSIM is adapted from W99 climatology, while that of SnowModel-LG is snow-free at the end of July 1979 and accumulates snow after that date. Meanwhile, compared with the CPOM product, the seasonal growth in SnowModel-LG is also larger, resulting in even higher April snow volume.

**The referee's comments 10:**

Line 331: "the inter-annual variability of monthly averaged..is small among" I don't get this. Small among all snow products? Small compared to what? What do you mean? Also it is difficult to see which points are November or April in Figure 4.

**Reply:**

Figure 6 is newly added and it serves the analysis of the seasonal cycle and interannual variability. The interannual variability here means for each snow product, the snow depth variability in November (or April) is calculated for the period of 2000-2018. The variabilities among all products are within 2.0cm in November (3.0cm in

April), which are smaller than the estimation of W99 climatology. Therefore, the authors states that these interannual variabilities are small.

***The referee's comments 11:***

*Line 335: "DuST show a significant positive. . ." where is the evidence?*

**Reply:**

Both Figure 3 and Figure 4 show consistent increasing snow depths from 2003-2008 (mean snow depth 16.0cm) to 2013-2018 (mean snow depth 20.0cm) period in DuST for October (autumn) and for March (late winter) (compared dashed and solid red lines). This is in contrast the reanalysis-based snow products.

***The referee's comments 12:***

*Line 337: "This features is also not. . ." where is the evidence?*

**Reply:**

According to Figure 5, there is no obviously trend in PMW Bremen nor in DESS during their respective period.

***The referee's comments 13:***

*Line 339: "We do not find. . ." Where can I see that? Figure 5 shows the trend from 1991 to 2015.*

**Reply:**

Consistent with the above trend analysis, there is no trend of Arctic mean snow depth in all products except DuST after year 2000 (Figure 5). Figure 7 reveals how snow change regionally from a longer time span.

***The referee's comments 14:***

*Line 353: "...directly fitted against OIB". What do you mean? You mean OIB data assimiled into those products?? ". . .show high correlations. . ." where is the number?*

**Reply:**

Based on data description in Section 2.3, the models for constructing snow depth estimations using DuST, PMW Bremen and DMI are directly trained/fitted from OIB products, which are not based on assimilation, however, directly rely on the specific OIB data used. The correlation values of OIB comparison are shown in Table 2.

***The referee's comments 15:***

*Line 357: "Figure 6. . ." Shouldn't this be first mentioned? Or Do you need this sentence? "The corresponding. . ." If you directly reference in the text, why do you need this?*

**Reply:**

The authors have rewritten the paragraph as: 'We assess the snow products against four different OIB snow depth products. We first compare OIB and snow products after gridding both to a common $100 \times 100$km grid and by evaluating the monthly averages in 2014 and 2015. Results are shown in Figure 10 and Table 2. Taking the quicklook product as an example, there are on average 1,300 OIB 40-m mean measurement samples per grid cell. It should be noted that snow depths from DuST, PMW Bremen and DMI are directly fitted against OIB snow depths, and as a result,

these data show high correlations (over 0.36 for PMW DMI) with OIB as shown in Table 2, which should not be taken as a real validation (or comparison) for these products. Except for NESOSIM and UW, other reanalysis-based products are also to some extent indirectly tuned by OIB snow depths in some years. In sum, all products show consistent high correlation with OIB, whereas the lowest correlation is seen for UW, which only correlates with some versions of OIB data (except quicklook).'

***The referee's comments 16:***

*Line 362: "Not surprisingly..." Why?*

**Reply:**

As mentioned before, although they use different versions of the OIB products to train the model, PMW Bremen and PMW DMI are directly fitted from OIB. Thus, their correlations are the highest when compared with OIB.

***The referee's comments 17:***

*Line 380: What is the key message here? Higher $R^2$ for coarse resolution but no significant difference for temperature resolution?*

**Reply:**

In order to avoid the potential problem of temporal and spatial averaging and interpolation during the validation, here the paper carries out the comparison for each of the data products on their native grids and native temporal resolutions. The results between monthly and daily scales hints that temporal resolution exerts only minor influences in the OIB comparison, since there are only small changes in $R^2$ and RMSE, without significant differences. For comparison, spatial resolution affects statistical fittings more, which is related to the lack of representation of OIB to these products at the coarse scale of 100km.

***The referee's comments 18:***

*Line 387: "Given. . ." it seems the results are sensitive to the choice of OIB data, but to me it does show some consistency, e.g., SnowModel-LG $R^2$ range from 0.27 to 0.47 yet PMW Bremen from 0.56 to 0.70. So, I don't know whether I should agree that it is impossible to conclude which ones perform better. Perhaps it is more related to how the particular products dependent to OIB data in their production?*

**Reply:**

It is indeed a bit confusing to have so many different OIB data products, each with different mean snow depth and spatial patterns. The authors acknowledge that for most products, the correlations with the various OIB products are over 0.2, except for UW. However, the comparison cannot be regarded as a validation for products that use OIB data to constrain the retrieved snow depths (i.e. PMW Bremen, PMW DMI and DuST). Furthermore, those which are indirectly fitted with OIB do not always have high correlations with the original version of the OIB dataset as used in their model development. For example, SnowModel-LG, DuST and PMW Bremen are based on OIB quicklook, while the best correlations are with the SRLD, quicklook and JPL OIB products respectively (although the differences are small). Although it is impossible to conclude which product is the best in this comparison, the paper does

find outliers among the products. Therefore, this paragraph is rewritten as: 'Given the potential data dependency problem and the sensitivity to the specific OIB data set, it is impossible to conclude which snow product performs best. Clearly snow products that have been produced through tuning with OIB data show higher $R^2$ and smaller RMSEs. The outlier is the UW product. In summary, there is a need for consensus as to which OIB data products are the most accurate, and also for further independent observations to compare against the various pan-Arctic snow products currently available to the science community.'

***The referee's comments 19:***

*Line 397: "PMW Bremen and. . ." no variability where I can see that? Figure 7? You mean no correlation? You*

*have Figure 7 as one of the main figure but very little description for that. What is the point showing this?*

**Reply:**

Note this is now Figure 11. No variability here means the spread of snow depth within the basin in PMW Bremen and PMW DMI is quite small compared to buoy data and other products. There is additionally no correlation against buoy measurements. The authors thank the referee suggestion and this paragraph is rewritten as: 'We further explore how well the snow products represent the temporal evolution of snow depth by comparing against CRREL IMBs and AWI snow buoys. As discussed in Section 2.1.1, 86 buoy tracks (58 tracks are from CRREL and 28 tracks are from AWI) are processed during the period of 2000 and 2017 (Table S1). It is worth noting that due to representation issues, we do not expect an exact match of any of the coarsely (i.e. 100km) resampled products to the local-scale of the buoy data. Therefore, the scatterplots in Figure 11 between monthly mean (March and April) buoy snow depths and those from the various products are based on their native spatial resolution. DuST is excluded due to lack of buoy samples in its more limited spatial coverage. Despite some statistically significant correlations, the correlations are all very low, with slopes close to 0. The highest correlation among the products is 0.16 for DESS. The PMW Bremen and PMW DMI products show essentially no variability/spread compared to the buoy data.'

***The referee's comments 20:***

*Line 400: I think it is far-fetched to compare each buoy with such products. I wonder why we see no*

*correlation.*

**Reply:**

This part of the analysis focuses on the snow accumulation process which are clearly an advantage of buoy measurements. The results do highlight the shortcomings of current products, and many factors may contribute to the zero correlations, including improper sea ice drift as used in reanalysis-based products. One thing to note is that, daily snow products have the potential in application of snow process investigation in the Arctic. Based on Stroeve et al. (2020), good correlations are witnessed between SnowModel-LG and buoy data following the buoy tracks in each integration step, while no/low correlation here may be the results of large discrepancies in trajectories determined from ice drift product and from buoy especially after a long-term (over three months) integration, which is also highlighted in the paper. And since the paper

cannot re-run each model to simulate snow changes along buoy true trajectories in each time step integration, thus purpose of validation with buoy here does not tend to pick the best product, but provide another potential validation view and more crucially, to state the importance of representation issue. **Therefore, the authors want to ask for the referee's suggestions about whether the buoy validation is suitable here.**

*The referee's comments 21:*

*Section 3.4: Personally, I like the results from this section, and found interesting among other results. The difficulty is that it is like comparing apple with an orange, but it does give us a general conclusion which is useful.*

**Reply:**

The authors have spitted the previous Section 3.4 and further reunited in Sec. 3.1-3.3 to differentiate current products and climatology in (1) mean state and distribution, (2) wintertime snow accumulation trend and (3) interannual variability and trend. The paper finds that (1) snow depth reduces from the 1980s to the 2000s which is consistent among the majority of products but with different amplitudes of reduction; (2) the flattened snow accumulation in W99 near the end of winter are quite different from in the current product; (3) the interannual variability in snow products is about half of previously reported in W99 climatology. All the above indicates that current snow conditions are quite different from the climatology. While a long-term increase of Arctic precipitation is expected in the future decades, it will more likely start to transition to liquid precipitation (Bintanja et al., 2014). Thus, how snow over sea ice changes and how it affects the ice and the climate as a whole are key scientific questions. The above comparisons with climatology are necessary and needed further discussion.

*The referee's comments 22:*

*5. Discussion: I found this is confusing about the scope of this research. Need to make it clear whether you want to show the inter-comparison results or developing a new sampling strategy. The whole discussions are about sampling strategy not about inter-comparison results. This actually tells me whether this is a research article or just for a discussion.*

**Reply:**

We thank the referee's comment on the scope of our study. The authors have moved the representation issues discussion into Section 4.3, following the OIB and buoy validation in Section 4.1 and 4.2. The purpose of this part is NOT to develop a sampling strategy, but to study how representation effects the validation works, especially regarding the low correlations with buoy measurements. The sampling strategies are designed to simulate various observational coverage (including buoy) and the representation's effects on validation, by using OIB dataset. Since this part serve as an important piece of evidence of validations with very limited snow depth measurements such as buoys, we consider it to be necessary. A full, systematic discussion of representation issue is beyond the scope of this paper, and planned as future work.

*6. Conclusions: It is too lengthy. Should be cut down to the key results and messages.*

**Reply:**

The authors have rewritten this part as Summary and Outlook. Some results have reduced in length in order to be more concise. The major conclusions are included. The choice of products that perform consistently during intercomparion and validation are stated explicitly, which answers the referee's comment on conveying key messages. Besides, extensions of the work in the future are also included.

**Reference**

Bintanja, Richard, and F. M. Selten. Future increases in Arctic precipitation linked to local evaporation and sea-ice retreat. Nature 509.7501 (2014): 479-482.

Kwok, R., Kacimi, S., Webster, M. A., Kurtz, N. T., & Petty, A. A. Arctic Snow Depth and Sea Ice Thickness From ICESat-2 and CryoSat-2 Freeboards: A First Examination. Journal of Geophysical Research: Oceans, 125(3), e2019JC016008, 2020.

Stroeve, J.C., Markus, T., Boisvert, L., Miller, J. and Barrett, A.: Changes in Arctic melt season and implications for sea ice loss. Geophysical Research Letters, 41(4), 1216-1225, 2014.

Stroeve, J., Liston, G. E., Buzzard, S., Zhou, L., Mallett, R., Barrett, A., Tschudi, M., M. Tsamados, P. I., and Stewart, J. S.: A Lagrangian snow-evolution system for sea-ice applications (SnowModel-LG): Part II - Analyses., Journal of Geophysical Research: Oceans, sumitted, 2020.

---

## Author Response (AR3)

**This content about the authors' response to 'Inter-comparison of snow depth over Arctic sea ice from reanalysis reconstructions and satellite retrieval' includes:**

(3) Marked-up manuscript version

The authors would like to thank the referee for the comments and suggestions to our manuscript, especially those involving quantitative results and analyses. We have revised the manuscript to the figures/histogram more clearly. Accordingly, we have made the following replies and the accompanying revisions.

***The referee's comments 1:***

*This is a second round review of the manuscript "Inter-comparison of snow depth over sea ice from multiple methods" submitted by Lu Zhou et al. The topic fits well to The Cryosphere and is of great and timely relevance. The authors have addressed most comments raised by the previous reviewers. I have some more specific and general suggestions.*

*The representation of the results is not always reproducible and accurate. In particular I am not satisfied with the graphical representation of the distributions. The underlying data are obscured by using kernel density smoothing for the histograms. It would be better to also show the underlying histograms. It is unclear how the averaging process of the gridding smears the distributions.*

*The graphical representation of scatter plots (Fig 10 and 13) is not very nice. The frequency of points is not visible. Better show density scatter plots like in https://doi.org/10.1029/2020GL088970*

**Reply:**

The authors acknowledge that the histogram could be obscured in the form of kernel density, however, we would like to argue that, by using kernel density function lines we do clearly and precisely show the differences of modes and variabilities among eight products (Figure 2). The following graph contains some products kernel density lines and histogram comparisons as in Figure 2.a. It is clear that the kernel density smooth lines convey the similar knowledge of the original histogram. Therefore, the authors choose kernel density smooth lines to better highlight the differences among the products (instead of raw histograms).

[Figure]

For the scatter plots in Figure 10, thank you for the suggestion. However, it is quite difficult to clearly differentiate the four pairs of comparison in one plot with the scatter density. The following figure is one example of comparing SnowModel-LG and OIB using the scatter densities, with different symbols representing different OIB products. It is quite hard to tell the differences among the four OIB product from the picture, although the linear fittings show remarkably different slopes. For Figure 13.b,

the authors have adopted scatter density to show the frequency of values.

[Figure]

***The referee's comments 2:***

*Numbers are reported not only with the significant digits which pretends too high accuracy, e.g. two modes of 18.0 cm and 32.0 cm. How can the mode be derived with subcentimeter accuracy? Units should not be set in italics.*

**Reply:**

The authors have changed the value in the accuracy of subcentimter and changed the units in non-italics.,

***The referee's comments 3:***

*The data availability (upon request from the authors) is against the FAIR principle which is adopted by the journal.*

**Reply:**

Thank you for pointing out this. Actually, in this intercomparison work we don't produce any new datasets. Meanwhile, all the public snow products, OIB and buoy data are explicitly covered in the data availability.

***The referee's comments 4:***

*The title is very general. It should include more details about the region (Arctic) and the methods.*

**Reply:**

The authors have changed the title into "Inter-comparison of snow depth over Arctic sea ice from reanalysis reconstructions and satellite retrieval".

***The referee's comments 5:***

*The abstract lacks details about the results. For example, I think the finding that the PMW Bremen and PMW DMI products show essentially no variability/spread compared to the buoy data is important to mention. I can not see the consistency in general structure among different snow depth data sets. Better remove this half-sentence from the abstract.*

**Reply:**

The abstract has been rewritten in the revision paper as following: 'In this study, we compare eight recently developed snow depth products over Arctic sea ice, which use satellite observations, modeling or a combination of satellite and modeling approaches. These products are further compared against various ground-truth observations, including those from ice mass balance observations and airborne measurements. Large mean snow depth discrepancies are observed over the Atlantic and Canadian Arctic sectors. There is no significant trend in the mean snow depth among all snow products since the 2000s, despite overall shallower snow packs in recent years compared to snow depth climatologies. The delaying in Arctic freeze-up could bring about the differences in early snow accumulation from products against the climatology. Among the products evaluated, the University of Washington (UW) snow depth product produces the deepest spring (March-April) snow packs, while the snow product from the Danish Meteorological Institute (DMI) provides the shallowest spring snow depths. SnowModel-LG and NESOSIM, also provide estimates of snow density and these two show no significant Arctic-wide trend during the 2000s.

Most snow products manifest the significant correlation with Operational IceBridge (OIB) while the correlations are quite low against buoy measurements, with no correlation and very low variability from University of Bremen and DMI products. Inconsistencies in reconstructed snow parameters among the products, as well as differences between in-situ and airborne observations can be partially attributed to differences in effective footprint and spatial/temporal coverage, as well as insufficient observations for validation/bias adjustments. Our results highlight the need for more targeted Arctic surveys over different spatial and temporal scales to allow for a more systematic comparison and fusion of airborne, in-situ and remote sensing observations.'

***The referee's comments 6:***

*Move supplementary plots into the main part if necessary for the understanding.*

**Reply:**

We've moved the original Figure 12 into Supplementary and exchanged the Figure S1, S2 and S3 into the manuscript.

The authors would like to thank the comments on specific term and suggestions on spatial aliasing to our revision. We have updated some figures in the main paper and the supplementary. As follows, we have made replies and the accompanying revisions.

*The referee's comments 1:*

*Review of Zhou, L., Stroeve, J., Xu, S., Petty, A., Tilling, R., Winstrup, M., Rostosky, P., Lawrence, I. R., Liston, G. E., Ridout, A., Tsamados, M., and Nandan, V.: Inter-comparison of snow depth over sea ice from multiple methods, The Cryosphere Discuss., https://doi.org/10.5194/tc-2020-65, in review, 2020.*

*This is a very timely paper on inter-comparison of snow-depth products derived using a variety of methods. Snow depth on sea ice is a hot research topic and this paper for the first time pulls together and inter-compares many different methods. The paper is very thorough in its comparisons and certainly deserves to be published asap. The paper has been reviewed already and I am only seeing this 2nd version, so I have only a few comments:*

*Section 2 and specifically 2.3.2: The term resolution is not used properly. On numerous occasions in the text, grid spacing is mis-interpreted as resolution. Please fix. Especially L250 and L266 but also L233-234. In products where low frequency PMR data are used, the spatial resolution is typically 30-50+ kilometers. Grid spacing can be much finer, but that does not mean resolution.*

**Reply:**

The authors thank reviewer for pointing out the difference between 'grid spacing' and 'resolution'. In Section 2, the products are described in different grids but not the resolution, especially for satellite-based products. The authors have revised the term "resolution" and "grid spacing" fits their situation and all revisions are marked out in red.

*The referee's comments 2:*

*L310-315: Discuss if the difference observed between reanalysis-based products over MY-ice can be explained by their different snow depth initialization procedures.*

**Reply:**

The differences among the reanalysis-based products especially over MYI are affected by many factors not only initial snow conditions, but also reanalysis datasets used, as well as the sea ice tracking algorithm they adopted. For SnowModel-LG, snowfall is from MERRA2, while NESOSIM uses the median value from MERRA, ERA-I, JRA55 and ASR reanalyses, and CPOM and UW both use ERA-I snowfall. The differences contributed from snow accumulation can be easily qualitative detected from the Eulerian perspective. The followed figure shows the differences between NESOSIM/CPOM products and NESOSIM/CPOM initial snow condition + SnowModel-LG accumulation (denoted as NESOSIM*/CPOM*) in March-April in 2015. Based on the same initial condition for NESOISM and CPOM, this differences directly manifest the large discrepancies in snow accumulation among SnowModel-LG, NESOSIM and CPOM, especially near the north of Greenland and the North Atlantic sector. Being consistent with Figure 7, CPOM shows overall less snow accumulation

than in SnowModel-LG. For revision, the authors would emphasize that the discrepancies among reanalysis-based products are the combined effect of several factors, on Line 295-300 of the manuscript and marked in red.

[Figure]

**The referee's comments 3:**

*L416: Note that SnowModel-LG has denser AND thinner snow than e.g. NESOSIM. This may correspond to very similar SWE. This observation could be included in this section.*

**Reply:**

The manuscripts mentioned the broadly equivalent SWE between SnowModel-LG and NESOSIM in Line 406, thank you for pointing out this.

**The referee's comments 4:**

*Section 4.3 You should also discuss and relate/refer to: Geiger, C., Müller, H., Samluk, J., Bernstein, E., & Richter-Menge, J. (2015). Impact of spatial aliasing on sea-ice thickness measurements. Annals of Glaciology, 56(69), 353-362. doi:10.3189/2015AoG69A644*

**Reply:**

The authors sincerely thank referee's suggestion and have added this discussion in Section 4.3, marked out in red.

**The referee's comments 5:**

*Just a suggestion: I would consider leaving out the comparison with buoy data since it does not bring any further insight. Alternatively move it to supplementary material and consider including some of the Suppl. Material in the main paper instead. Especially figures S1, S2 and S3 I believe should be in the main paper.*

**Reply:**

The authors would like to emphasize that, the comparison with buoy data highlights the importance of spatial representation problem, despite that the accumulation comparison brings about very limited information. Therefore, the snow accumulation comparison between buoy and the products (Figure 12) are moved into the Supplementary and accordingly, original figure S1, S2 and S3 are moved into the main manuscript.

**The referee's specific comments:**

*L20: observations. -> observations including the development of new products.*

----Corrected

*L39: lln -> ln*

----Corrected

*L118: within Arctic basin -> within the Artic basin*

----Corrected

*L173: consistence -> consistent*

----Corrected

*L325: snowpack is still significant -> the snowpack is still significantly*

----Corrected

*L343: discrepancies -> differences*

----Corrected

*L362-63: W99 snow accumulates -> W99 accumulates*

----Corrected

*L376: 2.0 cm to 3.0 cm -> 2.0 to 3.0 cm*

----Corrected

*L406: in 8 -> in Figure 8*

----Corrected

*L423: larger spatial scales -> coarser spatial scales*

----Corrected

*L442: "best against (R**2", some text seems to be missing here*

----Corrected

*L498: n -> ln*

----Corrected

*L539: percipitation -> precipitation*

----Corrected

**Inter-comparison of snow depth over Arctic sea ice from reanalysis reconstructions and satellite retrieval**

Lu Zhou[1], Julienne Stroeve[2,3,4], Shiming Xu[1,5], Alek Petty[6,7], Rachel Tilling[6,7], Mai Winstrup[8,9], Philip Rostosky[10], Isobel R. Lawrence[11], Glen E. Liston[12], Andy Ridout[2], Michel Tsamados[2], and Vishnu Nandan[3]

[1]Ministry of Education Key Laboratory for Earth System Modeling, Department of Earth System Science, Tsinghua University, Beijing, China
[2]Centre for Polar Observation and Modelling, Earth Sciences, University College London, London, UK
[3]Centre for Earth Observation Science, University of Manitoba, Winnipeg, Canada
[4]National Snow and Ice Data Center, University of Colorado, Boulder, CO, USA
[5]University Corporation for Polar Research, Beijing, China
[6]NASA Goddard Space Flight Center, Greenbelt, MD, USA
[7]Earth System Science Interdisciplinary Center, University of Maryland, College Park, MD, USA
[8]DTU Space, Technical University of Denmark, Lyngby, Denmark
[9]Danish Meteorological Institute (DMI), Copenhagen, Denmark
[10]University of Bremen, Institute of Environmental Physics, Bremen, Germany
[11]Centre for Polar Observation and Modelling, University of Leeds, UK
[12]Colorado State University, Cooperative Institute for Research in the Atmosphere (CIRA), Fort Collins, CO, USA

**Correspondence:** Julienne Stroeve (stroeve@nsidc.org)

**Abstract.**

In this study, we compare eight recently developed snow depth products over Arctic sea ice, which use satellite observations, modeling or a combination of satellite and modeling approaches. These products are further compared against various ground-truth observations, including those from ice mass balance observations and airborne measurements. Large mean snow depth discrepancies are observed over the Atlantic and Canadian Arctic sectors. The differences between climatology and the snow products early in winter could be in part a result of the delaying in Arctic ice formation that reduces early snow accumulation. Besides, late winter snow accumulation rate is quite slower in the climatology than in the newly products in spite of the overall shallower spring snow among all products. 
[revised manuscript text omitted]